# Contextual Linear Bandits with Delay as Payoff

**Mengxiao Zhang** [1]  **Yingfei Wang** [2]  **Haipeng Luo** [3]

## Abstract

A recent work by Schlisselberg et al. (2025) studies a delay-as-payoff model for stochastic multi-armed bandits, where the payoff (either loss or reward) is delayed for a period that is proportional to the payoff itself. While this captures many real-world applications, the simple multi-armed bandit setting limits the practicality of their results. In this paper, we address this limitation by studying the delay-as-payoff model for contextual linear bandits. Specifically, we start from the case with a fixed action set and propose an efficient algorithm whose regret overhead compared to the standard no-delay case is at most $D\Delta_{\max} \log T$, where $T$ is the total horizon, $D$ is the maximum delay, and $\Delta_{\max}$ is the maximum suboptimality gap. When payoff is loss, we also show further improvement of the bound, demonstrating a separation between reward and loss similar to Schlisselberg et al. (2025). Contrary to standard linear bandit algorithms that construct least squares estimator and confidence ellipsoid, the main novelty of our algorithm is to apply a phased arm elimination procedure by only picking actions in a *volumetric spanner* of the action set, which addresses challenges arising from both payoff-dependent delays and large action sets. We further extend our results to the case with varying action sets by adopting the reduction from Hanna et al. (2023). Finally, we implement our algorithm and showcase its effectiveness and superior performance in experiments.

## 1. Introduction

Stochastic multi-armed bandit (MAB) is a well-studied theoretical framework for sequential decision making. In recent years, considerable investigation has been given to the realistic situations where the agent observes the payoff (either reward or loss) of an arm only after a certain delayed period of time. However, most work assumes that the delays are *payoff-independent*. Namely, while the delay may depend on the chosen arm, it is sampled independently from the stochastic payoff of the chosen arm.

Lancewicki et al. (2021) address this limitation by studying a setting where the delay and the reward are drawn together from a joint distribution. Later, Tang et al. (2024) consider a special case where the delay is exactly the reward. Their motivation stems from response-adaptive clinical trials that aim at maximizing survival outcomes. For example, progression-free survival (PFS)—defined as the number of days after treatment until disease progression or death—is widely used to evaluate the effectiveness of a treatment. Notably, in this context, the "delays" in observing the PFS are the PFS itself. Schlisselberg et al. (2025) build on and refine this investigation, extending the study to the case where delay is the loss itself. Taken together, this delay-as-payoff framework effectively captures many real-world scenarios involving time-to-event data across many domains. For example, postoperative length of stay (PLOS) is one example of time-to-event data that specifies the length of stay after surgery. Potential surgical procedures and postoperative care can be modeled as arms. The delay—defined as the time until the patient is discharged—can be interpreted as the loss that we aim to minimize. As another example, in advertising, common metrics, including Average Time on Page (ATP) and Time to Re-engagement (that tracks the time elapsed between a user's initial interaction with an ad and subsequent engagement such as returning to the website), can be modeled as reward or loss inherently delayed by the same duration.

Despite such recent progress, the current consideration of *payoff-dependent* delay remains limited to the simple multi-armed bandit (MAB) setting. While MAB frameworks are foundational in decision-making problems, they have notable practical limitations. Specifically, they fail to account for the influence of covariates that drive heterogeneous responses across different actions. This makes them less suitable for scenarios involving a large number of (potentially dynamically changing) actions and/or situations where context is crucial in shaping outcomes.

---

[1]University of Iowa [2]University of Washington [3]University of Southern California. Correspondence to: Mengxiao Zhang <mengxiao-zhang@uiowa.edu>.

*Proceedings of the $42^{nd}$ International Conference on Machine Learning*, Vancouver, Canada. PMLR 267, 2025. Copyright 2025 by the author(s).

**Contributions.** Motivated by this limitation, in this work, we extend the delay-as-payoff model from MAB to contextual linear bandits, a practical framework that is widely used in real-world applications. Specifically, our contributions are as follows.

- As a first step, in Section 3, we study stochastic linear bandits with a fixed action set (known as the non-contextual setting). We point out the difficulty of directly combining the standard LinUCB algorithm with the idea of Schlisselberg et al. (2025), and propose a novel phased arm elimination algorithm that only selects actions from a *volumetric spanner* of the action set. In the delay-as-loss case, we prove that, compared to the standard regret in the delay-free setting, the overhead caused by the payoff-dependent delay for our algorithm is $\mathcal{O}(\min\{nd^\star \log(T/n) + D\Delta_{\max}, D\Delta_{\max} \log(T/n)\})$, where $n$ is the dimension of the action set, $T$ is the total horizon, $\Delta_{\max}$ is the maximum suboptimality gap, $d^\star$ is the expected delay of the optimal action, and $D$ is the maximum possible delay (formal definitions are deferred to Section 2). This instance-dependent bound is in the same spirit as the one of Schlisselberg et al. (2025) and is small whenever the optimal action has a small loss. In the delay-as-reward case, a slightly worse bound is provided in Appendix B; such a separation between loss and reward is similar to the results of Schlisselberg et al. (2025).

- Next, in Section 4, we extend our results to the contextual case where the action set is varying and drawn i.i.d. from an unknown distribution. Using a variant of our non-contextual algorithm (that can handle misspecification) as a subroutine, we apply the contextual to non-contextual reduction recently proposed by (Hanna et al., 2023) and show that the resulting algorithm enjoys a similar regret guarantee despite having varying action sets, establishing the first regret guarantee for contextual linear bandits with delay-as-payoff.

- In Section 5, we implement our algorithm and test it on synthetic linear bandits instances, demonstrating its superior performance against a baseline that runs LinUCB with only the currently available feedback.

**Related works.** Recent research has investigated different problems of learning under bandit feedback with delayed payoff, addressing various new challenges caused by the combination of delay and bandit feedback. As mentioned, most studies assume *payoff-independent* delays. Among this line of research, Dudík et al. (2011) are among the first to consider delays in stochastic MAB with a constant delay. Mandel et al. (2015) and Joulani et al. (2013) extend the consideration to stochastic delays, with the assumption that the delay is bounded.

Subsequent studies on i.i.d. stochastic delays differentiate between *arm-independent* and *arm-dependent* delays. For *arm-independent* delays, Zhou et al. (2019); Vernade et al. (2020a); Blanchet et al. (2024) show regret characterizations for (generalized) linear stochastic contextual bandits. Pike-Burke et al. (2018) consider aggregated anonymous feedback, under the assumption that the expected delay is bounded and known to the learner. *Arm-dependent* stochastic delays have been explored in various settings, including Gael et al. (2020); Arya & Yang (2020); Lancewicki et al. (2021).

Far less attention has been given to *payoff-dependent* stochastic delays. The setting in Vernade et al. (2017) implies a dependency between the reward and the delay, as a current non-conversion could be the result of a delayed reward of 1. Lancewicki et al. (2021) consider the case where the stochastic delay in each round and the reward are drawn from a general joint distribution. Tang et al. (2024) investigate strongly reward-dependent delays, specifically motivated by medical settings where the delay is equal to the reward. Schlisselberg et al. (2025) follow this investigation and extend the discussion to delay as loss, and provide a tighter regret bound. Although with a slightly different focus, Thune et al. (2019); Zimmert & Seldin (2020); Gyorgy & Joulani (2021); Van Der Hoeven & Cesa-Bianchi (2022); van der Hoeven et al. (2023) and several other works study non-stochastic bandits, where both the delay and rewards are adversarial.

Nevertheless, the *payoff-dependent* (either loss or reward) delays are only studied under stochastic multi-armed bandits (MAB). In this work, we extend the study to contextual linear bandits, significantly broadening its practicality.

## 2. Preliminary

Throughout this paper, we use $[N]$ to denote $\{1, 2, \ldots, N\}$ for some positive integer $N$. Let $\mathbb{R}_+^n$ be the $n$-dimensional Euclidean space in the positive orthant and $\mathbb{B}_2^n(1) = \{v \in \mathbb{R}^n : \|v\|_2 \leq 1\}$ be the $n$-dimensional unit ball with respect to $\ell_2$ norm. Define $\mathcal{U}[a, b]$ to be the uniform distribution over $[a, b]$. For a real number $a$, define $\mathrm{SGN}(a)$ as the sign of $a$. For two real numbers $a$ and $b$, define $a \vee b \triangleq \max\{a, b\}$. For a finite set $\mathcal{S}$, denote $|\mathcal{S}|$ as the cardinality of $\mathcal{S}$. The notation $\widetilde{\mathcal{O}}(\cdot)$ hides all logarithmic dependencies.

In this paper, we consider the delay-as-payoff model proposed by Schlisselberg et al. (2025), in which the delay of the payoff is proportional to the payoff itself. Specifically, we study stochastic linear bandits in this model, and we start with a fixed action set as the first step (referred to as the non-contextual case) and then move on to the case with a time-varying action set (referred to as the contextual case). For conciseness, we mainly discuss the payoff-as-loss case,

but our algorithm and results can be directly extended to the payoff-as-reward case (see Appendix B).

**Non-contextual stochastic linear bandits.** In this problem, the learner is first given a *fixed* finite set of actions $\mathcal{A} \subset \mathbb{R}_+^n \cap \mathbb{B}_2^n(1)$ with cardinality $|\mathcal{A}| = K$. Let $D > 0$ be the maximum possible delay. At each round $t \in [T]$, the learner selects an action $a_t \in \mathcal{A}$ and incurs a loss $u_t = \mu_{a_t} + \eta_t \in [0, 1]$ where $\eta_t$ is zero-mean random noise, $\mu_a = \langle a, \theta \rangle$ is the expected payoff of action $a$, and $\theta \in \mathbb{R}_+^n \cap \mathbb{B}_2^n(1)$ is the model parameter that is unknown to the learner.[1] Then, the loss is received by the learner at the end of round $\lceil t + d_t \rceil$ where the delay $d_t = D \cdot u_t$ (that is, proportional to the loss). The goal of the learner is to minimize the (expected) pseudo regret defined as follows:

$$\mathrm{Reg} \triangleq \mathbb{E} \left[ \sum_{t=1}^{T} \langle a_t, \theta \rangle \right] - T \cdot \min_{a \in \mathcal{A}} \langle a, \theta \rangle. \quad (1)$$

Let $a^\star \in \mathrm{argmin}_{a \in \mathcal{A}} \langle a, \theta \rangle$ be an optimal action, $\mu^\star = \mu_{a^\star}$ be its expected loss, and $d^\star = D\mu^\star$ be its expected delay. For an action $a$, define $\Delta_a = \langle a - a^\star, \theta \rangle$ as its sub-optimality gap. Further define $\Delta_{\min} = \min_{a \in \mathcal{A}, \Delta_a > 0} \Delta_a$ and $\Delta_{\max} = \max_{a \in \mathcal{A}} \Delta_a$ to be the minimum and maximum non-zero sub-optimality gap respectively.

We point out that the standard multi-armed bandit (MAB) setting considered in Schlisselberg et al. (2025) is a special case of our setting with $\mathcal{A}$ being the set of all standard basis vectors in $\mathbb{R}^n$.

**Contextual stochastic linear bandits.** In the contextual case, the main difference is that the action set is not fixed but *changing over rounds*. Specifically, at each round $t$, the learner first receives an action set $\mathcal{A}_t \subset \mathbb{R}_+^n \cap \mathbb{B}_2^n(1)$ (which can be seen as a context), where we assume that $\mathcal{A}_t$ is drawn i.i.d. from an unknown distribution $\mathcal{P}$. The rest of the protocol remains the same, and the goal of the learner is still to minimize the (expected) pseudo regret, defined as:

$$\mathrm{Reg} \triangleq \mathbb{E} \left[ \sum_{t=1}^{T} \langle a_t, \theta \rangle - \sum_{t=1}^{T} \min_{a_t^\star \in \mathcal{A}_t} \langle a_t^\star, \theta \rangle \right],$$

where the expectation is taken over both the internal randomness of the algorithm and the external randomness in the action sets and loss noises.

## 3. First Step: Non-Contextual Linear Bandits

In this section, we focus on the non-contextual case, which serves as a building block for eventually solving the contextual case. Before introducing our algorithm, we first briefly introduce the successive arm elimination algorithm

for the simpler MAB setting proposed by Schlisselberg et al. (2025) and their ideas of handling payoff-dependent delay. Specifically, their algorithm starts with a guess $B = 1/D$ on the optimal action's loss, and maintains an active set of arms. The algorithm pulls each arm in the active set once, and constructs two LCB's (lower confidence bound) and one UCB (upper confidence bound) for each action in the active set as follows (supposing the current round being $t$):

$$\mathrm{LCB}_{t,1}(a) = \frac{1}{k_t(a)} \sum_{\tau \in O_t(a)} u_\tau - \sqrt{\frac{2 \log T}{k_t(a)}}, \quad (2)$$

$$\mathrm{LCB}_{t,2}(a) = \frac{1}{c_t(a)} \sum_{\tau \in C_t(a)} u_\tau - \sqrt{\frac{2 \log T}{c_t(a) \vee 1}}, \quad (3)$$

$$\mathrm{UCB}_t(a) = \frac{1}{c_t(a)} \sum_{\tau \in C_t(a)} u_\tau + \sqrt{\frac{2 \log T}{c_t(a) \vee 1}}, \quad (4)$$

where $k_t(a) = \sum_{\tau=1}^{t} \mathbb{1}\{a_t = a\}$ is the total number of pulls of action $a$ till round $t$, $O_t(a) = \{\tau : \tau + d_\tau \leq t \text{ and } a_\tau = a\}$ is the set of rounds where action $a$ is chosen and its loss has been received by the end of round $t$, $C_t(a) = \{\tau \leq t - D : a_\tau = a\}$ is the set of rounds up to $t - D$ where action $a$ is chosen (so its loss has for sure been received by the end of round $t$), and $c_t(a) = |C_t(a)|$. Specifically, Eq. (2) constructs an LCB of action $a$ assuming all the action's unobserved loss to be $0$ (the smallest possible), while Eq. (3) and Eq. (4) construct an LCB and a UCB using only the losses no later than round $t - D$ (which must have been received by round $t$), making the empirical average a better estimate of the expected loss. With $\mathrm{UCB}_t(a)$ and $\mathrm{LCB}_t(a) = \max\{\mathrm{LCB}_{t,1}(a), \mathrm{LCB}_{t,2}(a)\}$ constructed, the algorithm eliminates an action $a$ if its $\mathrm{LCB}_t(a)$ is larger than $\min\{\mathrm{UCB}_t(a'), B\}$ for some $a'$ in the active set. If all the actions are eliminated, this means that the guess $B$ on the optimal loss is too small, and the algorithm starts a new epoch with $B$ doubled.[2]

**Challenges** However, this approach cannot be directly applied to linear bandits. Specifically, standard algorithms for stochastic linear bandits without delay (e.g., Li et al. (2010); Abbasi-Yadkori et al. (2011)) all construct UCB/LCB for each action by constructing an ellipsoidal confidence set for $\theta$. In the delay-as-payoff model, while it is still viable to construct UCB/LCB similar to Eq. (3) and Eq. (4) via a standard confidence set of $\theta$, it is difficult to construct an LCB counterpart similar to Eq. (2). This is because one action's loss is estimated using observations of all other actions in linear bandits, and naively treating the unobserved

---

[1]We enforce both $\mathcal{A} \subset \mathbb{R}_+^n \cap \mathbb{B}_2^n(1)$ and $\theta \in \mathbb{R}_+^n \cap \mathbb{B}_2^n(1)$ to make sure that the payoff (and hence the delay) is non-negative.

[2]In fact, Schlisselberg et al. (2025) construct yet another LCB based on the number of unobserved losses. We omit this detail since we are not able to use this to further improve our bounds for linear bandits.

loss of one action as zero might not necessarily lead to an underestimation of another action.

**Our ideas** To bypass this barrier, we give up on estimating $\theta$ itself and propose to construct UCB/LCB for each action using the observed losses of the *volumatric spanner* of the action set. A volumetric spanner of an action set $\mathcal{A}$ is defined such that every action in $\mathcal{A}$ can be expressed as a linear combination of the spanner.

**Definition 3.1** (Volumetric Spanner (Hazan & Karnin, 2016))**.** Suppose that $\mathcal{A} = \{a_1, a_2, \ldots, a_N\}$ is a set of vectors in $\mathbb{R}^n$. We say $\mathcal{S} \subseteq \mathcal{A}$ is a *volumetric spanner* of $\mathcal{A}$ if for any $a \in \mathcal{A}$, we can write it as $a = \sum_{b \in \mathcal{S}} \lambda_b \cdot b$ for some $\lambda \in \mathbb{R}^{|\mathcal{S}|}$ with $\|\lambda\|_2 \le 1$.

Due to the linear structure, it is clear that the loss $\mu_a$ of action $a$ can be decomposed in a similar way as $\sum_{b \in \mathcal{S}} \lambda_b \mu_b$, making it possible to estimate every action's loss by only estimating the loss of the spanner. Moreover, such a spanner can be efficiently computed:

**Proposition 3.2** (Bhaskara et al. (2023))**.** *Given a finite set $\mathcal{A}$ of size $K$, there exists an efficient algorithm finding a volumetric spanner $\mathcal{S}$ of $\mathcal{A}$ with $|\mathcal{S}| = 3n$ within $\mathcal{O}(Kn^3 \log n)$ runtime.*

Equipped with the concept of volumetric spanner, we are now ready to introduce our algorithm (see Algorithm 1). Specifically, our algorithm also makes a guess $B$ on the loss of the optimal action. With this guess, it proceeds to multiple epochs of arm elimination procedures, with the active action set initialized as $\mathcal{A}_1 = \mathcal{A}$. In each epoch $m$, instead of picking every action in the active set $\mathcal{A}_m$, we first compute a volumetric spanner $\mathcal{S}_m$ of $\mathcal{A}_m$ with $|\mathcal{S}_m| = 3n$ (Line 4), which can be done efficiently according to Proposition 3.2, and then pick each action in the spanner set $\mathcal{S}_m$ for $2^m$ rounds in a round-robin way (Line 5).

After that, we calculate two UCBs and two LCBs for actions in the spanner, in a way similar to the simpler MAB setting discussed earlier (Line 7). Specifically, the first one is in the same spirit of Eq. (2): we calculate $\widehat{\mu}_m^+(a)$ ($\widehat{\mu}_m^-(a)$) as an overestimation (underestimation) of the expected loss of action $a$ by averaging over all observed losses from the rounds in $O_m(a)$ as well as the maximum (minimum) possible value of unobserved losses from the rounds in $E_m(a)$; see Eq. (5) and Eq. (6). The first UCB (LCB) $\widehat{\mu}_{m,1}^+(a)$ ($\widehat{\mu}_{m,1}^-(a)$) is then computed based on $\widehat{\mu}_m^+(a)$ ($\widehat{\mu}_m^-(a)$) by incorporating a standard confidence width $\frac{\beta}{\sqrt{2^m}}\|a\|_2$ for some coefficient $\beta$; see Eq. (7) and Eq. (8). Then, to compute the second UCB/LCB, which is in the same spirit as Eq. (3) and Eq. (4), we calculate an unbiased estimation $\widehat{\mu}_m^F(a)$ of the expected loss of $a$ by averaging losses from the rounds in $C_m(a)$, that is, all the rounds where the observation must have been revealed; see Eq. (9). Note that the number of

---

**Algorithm 1:** Phased Elimination via Volumetric Spanner for Linear Bandits with Delay-as-Loss

1 Input: maximum possible delay $D$, action set $\mathcal{A}$, $\beta > 0$.
2 Initialization: optimal loss guess $B = 1/D$.
3 Initialization: active action set $\mathcal{A}_1 = \mathcal{A}$.
**for** $m = 1, 2, \ldots,$ **do**
4     Find $\mathcal{S}_m = \{a_{m,1}, \ldots, a_{m,|\mathcal{S}_m|}\}$, a volumetric spanner of $\mathcal{A}_m$ with $|\mathcal{S}_m| = 3n$.
5     Pick each $a \in \mathcal{S}_m$ $2^m$ times in a round-robin way.
6     Let $\mathcal{I}_m$ contain all the rounds in this epoch.
7     For each $a \in \mathcal{S}_m$, calculate the following quantities:

$$\widehat{\mu}_m^+(a) = \frac{1}{2^m}\Big(\sum_{\tau \in O_m(a)} u_\tau + \sum_{\tau \in E_m(a)} 1\Big), \quad (5)$$

$$\widehat{\mu}_m^-(a) = \frac{1}{2^m}\sum_{\tau \in O_m(a)} u_\tau, \quad (6)$$

$$\widehat{\mu}_{m,1}^+(a) = \widehat{\mu}_m^+(a) + \frac{\beta}{2^{m/2}}\|a\|_2, \quad (7)$$

$$\widehat{\mu}_{m,1}^-(a) = \widehat{\mu}_m^-(a) - \frac{\beta}{2^{m/2}}\|a\|_2, \quad (8)$$

$$\widehat{\mu}_m^F(a) = \frac{1}{c_m(a)}\sum_{\tau \in C_m(a)} u_\tau, \quad (9)$$

$$\widehat{\mu}_{m,2}^+(a) = \widehat{\mu}_m^F(a) + \frac{\beta}{\sqrt{c_m(a)}}\|a\|_2, \quad (10)$$

$$\widehat{\mu}_{m,2}^-(a) = \widehat{\mu}_m^F(a) - \frac{\beta}{\sqrt{c_m(a)}}\|a\|_2, \quad (11)$$

    where $c_m(a) = |C_m(a)|$,
    $C_m(a) = \{\tau \in \mathcal{I}_m : \tau + D \in \mathcal{I}_m, a_\tau = a\}$,
    $O_m(a) = \{\tau \in \mathcal{I}_m : \tau + d_\tau \in \mathcal{I}_m, a_\tau = a\}$, and
    $E_m(a) = \{\tau \in \mathcal{I}_m : a_\tau = a\} \setminus O_m(a)$.
**for** *each* $a \in \mathcal{A}_m$ **do**
8         Decompose $a$ as $a = \sum_{i=1}^{|\mathcal{S}_m|} \lambda_{m,i}^{(a)} a_{m,i}$ with $\|\lambda_m^{(a)}\|_2 \le 1$ and calculate

$$\text{UCB}_m(a) = \sum_{i=1}^{|\mathcal{S}_m|} \lambda_{m,i}^{(a)} \cdot \widehat{\mu}_{m,2}^{\text{SGN}(\lambda_{m,i}^{(a)})}(a_{m,i}),$$
$$(12)$$

$$\text{LCB}_m(a) = \max_{j \in \{1,2\}}\{\text{LCB}_{m,j}(a)\} \text{ where}$$

$$\text{LCB}_{m,j}(a) = \sum_{i=1}^{|\mathcal{S}_m|} \lambda_{m,i}^{(a)} \cdot \widehat{\mu}_{m,j}^{\text{SGN}(-\lambda_{m,i}^{(a)})}(a_{m,i}),$$
$$(13)$$

9     Set $\mathcal{A}_{m+1} = \mathcal{A}_m$.
**for** $a \in \mathcal{A}_m$ **do**
10         **if** $\exists a' \in \mathcal{A}_m$, *s.t.*
        $\text{LCB}_m(a) \ge \min\{\text{UCB}_m(a'), B\}$ **then**
            Eliminate $a$ from $\mathcal{A}_{m+1}$.
11     **if** $\mathcal{A}_{m+1} = \emptyset$ **then**
        Set $B \leftarrow 2B$ and go to Line 3.

---

such rounds, $c_m(a) = |C_m(a)|$, is a fixed number, and thus $\widehat{\mu}_m^F(a)$ is indeed unbiased. Similarly, we incorporate a standard confidence width $\frac{\beta}{\sqrt{c_m(a)}} \|a\|_2$ to arrive at the second UCB $\widehat{\mu}_{m,2}^+(a)$ and LCB $\widehat{\mu}_{m,2}^-(a)$; see Eq. (10) and Eq. (11).

The next step of our algorithm is to use these UCBs/LCBs for the spanner to compute corresponding UCBs/LCBs for every active action in $\mathcal{A}_m$ (Line 8). Specifically, for each action $a \in \mathcal{A}_m$, according to the definition of a volumetric spanner (Definition 3.1), we can write $a$ as a linear combination of actions in $\mathcal{S}_m$: $\sum_{i=1}^{|S_m|} \lambda_{m,i}^{(a)} a_{m,i}$. As mentioned, due to the linear structure of losses, we also have $\mu_a = \sum_{i=1}^{|S_m|} \lambda_{m,i}^{(a)} \mu_{a_{m,i}}$. Thus, when constructing a UCB (or similarly LCB) for $a$, based on whether $\lambda_{m,i}^{(a)}$ is positive or not, we decide whether to use the UCB or LCB of $a_{m,i}$; see Eq. (12), a counterpart of Eq. (4), and Eq. (13), a counterpart of Eq. (2) and Eq. (3).[3]

At the end of an epoch, we eliminate all actions from the active action set if their LCB is either larger than the UCB of certain action or the guess $B$ on the optimal loss (Line 10). If the active set becomes empty, this means that the guess $B$ is too small, and the algorithm restarts with the guess doubled; otherwise, we continue to the next epoch.

**Computational complexity**  We analyze the computational complexity of Algorithm 1. Specifically, the total runtime of our algorithm over $T$ rounds is $\mathcal{O}(nT + Kn^3 \log n \log(T/n))$, as we compute the volumetric spanner only once per epoch, and the total number of epochs is $\mathcal{O}(\log(T/n))$. Compared to the classic LinUCB algorithm (Abbasi-Yadkori et al., 2011), our approach is in fact more computationally efficient: LinUCB computes the UCB for each action at a cost of $\mathcal{O}(n^2)$ per round, resulting in a total runtime of $\mathcal{O}(Kn^2 T)$.

**Theoretical performance**  We prove the following regret bound for our algorithm.

**Theorem 3.3.** *Algorithm 1 with $\beta = \sqrt{2 \log(KT^3)}$ guarantees:*

$$\mathrm{Reg} \leq \mathcal{O}\left(\min\{V_1, V_2\}\right) + \log(d^\star) \cdot \mathcal{O}\left(\min\{W_1, W_2\}\right),$$

*where* $V_1 = \frac{n^2 \log(KT) \log(T/n) \log(d^\star)}{\Delta_{\min}}$, $V_2 = n\sqrt{T \log(d^\star) \log(KT)}$, $W_1 = nd^\star \log(T/n) + D\Delta_{\max}$, *and* $W_2 = D\Delta_{\max} \log(T/n)$.

The first term in the regret bound $\mathcal{O}\left(\min\{V_1, V_2\}\right)$ is of order $\widetilde{\mathcal{O}}(\min\{\frac{n^2}{\Delta_{\min}}, n\sqrt{T}\})$, which matches the standard regret bound of LinUCB in the case without delay (Abbasi-Yadkori et al., 2011). The second term is the overhead

[3]This also explains why we need $\widehat{\mu}_m^+(a)$, a quantity not used in Schlisselberg et al. (2025).

caused by delay and is in the same spirit as the result of Schlisselberg et al. (2025): focusing only on the part that grows in $T$, we see that $W_1$ only depends on $d^\star$, the expected delay of the optimal action (and hence the smallest delay among all actions), while $W_2$ depends on the maximum possible delay $D$ but scaled by $\Delta_{\max}$, the largest sub-optimality gap. Therefore, the delay overhead of our algorithm is small when either the shortest delay is small or all actions have similar losses. We remark again that in the delay-as-reward setting, we obtain similar results; see Appendix B for details.

### 3.1. Analysis

In this section, we provide a proof sketch of Theorem 3.3. Detailed proofs are deferred to Appendix A.

The proof starts by proving that $\mathrm{UCB}_m(a)$ and $\mathrm{LCB}_m(a)$ are indeed valid UCB and LCB respectively for all actions in $\mathcal{A}_m$. This follows from first using standard concentration inequalities to show that $\widehat{\mu}_{m,1}^+(a)$ and $\widehat{\mu}_{m,2}^+(a)$ ($\widehat{\mu}_{m,1}^-(a)$ and $\widehat{\mu}_{m,2}^-(a)$) are valid UCBs (LCBs) for each action in the spanner, and then generalizing it to every action $a \in \mathcal{A}_m$ according to its decomposition over the actions in the spanner.

With this property, our analysis then proceeds to control the regret of Algorithm 1 for each guess of $B$ separately. Let $\mathcal{T}_B$ be the set of rounds when Algorithm 1 runs with guess $B$. In **Step 1**, we first show that the use of $\mathrm{LCB}_{m,2}(a)$ and $\mathrm{UCB}_m(a)$ ensures a regret bound of $\mathcal{O}\left(\min\{R_1, R_2\} + D\Delta_{\max} \log(T/n)\right)$ where $R_1 = \frac{n^2 \log(KT) \log(T/n)}{\Delta_{\min}}$ and $R_2 = n\sqrt{|\mathcal{T}_B| \log(KT)}$, and then in **Step 2**, we show that the use of $\mathrm{LCB}_{m,1}(a)$ and $\mathrm{UCB}_m(a)$ ensures a regret bound of $\mathcal{O}(\min\{R_1, R_2\} + (nd^\star + DB) \log(T/n) + D\Delta_{\max})$.

**Step 1**  For notational convenience, we define

$$\mathsf{rad}_{m,a}^F = \beta \sum_{i=1}^{|\mathcal{S}_m|} |\lambda_{m,i}^{(a)}| \cdot \frac{\|a\|_2}{\sqrt{c_m(a_{m,i})}}$$

to be the total confidence radius of action $a$ coming from the definition of $\mathrm{LCB}_{m,2}(a)$ and $\mathrm{UCB}_m(a)$. Via a standard analysis of arm elimination, we show that that if an action $a$ is not eliminated at the end of epoch $m$, we have

$$\Delta_a \leq 4 \max_{a \in \mathcal{A}_m} \mathsf{rad}_{m,a}^F \leq \frac{4\sqrt{3n}\beta}{\min_{a_m \in \mathcal{S}_m} \sqrt{c_m(a_m)}},$$

where the second inequality uses Cauchy-Schwarz inequality and the properties of volumetric spanners, specifically that $\|\lambda_m^{(a)}\|_2 \leq 1$ and $|\mathcal{S}_m| = 3n$. To provide a lower bound on $c_m(a')$ for any $a' \in \mathcal{S}_m$, note that we pick each action

$a' \in \mathcal{S}_m$ $2^m$ times in a round-robin manner, and thus

$$c_m(a') \geq 2^m - \frac{D}{|\mathcal{S}_m|} - 1 = 2^m - \frac{D}{3n} - 1.$$

Rearranging the terms, we then obtain

$$2^m \Delta_a \leq \frac{48n\beta^2}{\Delta_a} + \frac{D\Delta_a}{3n} + \Delta_a. \tag{14}$$

Taking summation over all $a \in \mathcal{S}_m$ and $m$, and noticing that the total number of epochs is bounded by $M = \lceil \log_2(|\mathcal{T}_B|/3n) \rceil$, we arrive at the following $\mathcal{O}(R_1 + D\Delta_{\max} \log(T/n))$ regret guarantee:

$$\sum_{m=1}^{M} \sum_{a \in \mathcal{S}_m} 2^m \Delta_a$$

$$\leq \sum_{m=1}^{M} \sum_{a \in \mathcal{S}_m, \Delta_a > 0} 2 \cdot \left( \frac{48n\beta^2}{\Delta_a} + \frac{D\Delta_a}{3n} + \Delta_a \right)$$

$$\leq \sum_{m=1}^{M} \sum_{a \in \mathcal{S}_m, \Delta_a > 0} \mathcal{O}\left( \frac{n \log(KT)}{\Delta_a} \right) + \mathcal{O}\left( D\Delta_{\max} \log(T/n) \right),$$

$$\leq \mathcal{O}\left( \frac{n^2 \log(T/n) \log(KT)}{\Delta_{\min}} \right) + \mathcal{O}\left( D\Delta_{\max} \log(T/n) \right),$$

where the first inequality is because $a \in \mathcal{S}_m$ is not eliminated in epoch $m-1$ and the last inequality is by lower bounding $\Delta_a$ by $\Delta_{\min}$.

To obtain the other instance-independent regret bound $\mathcal{O}(R_2 + D\Delta_{\max} \log(T/n))$, we bound the regret differently by considering $\Delta_a \geq \beta\sqrt{n/2^m}$ and $\Delta_a \leq \beta\sqrt{n/2^m}$ separately:

$$\sum_{m=1}^{M} \sum_{a \in \mathcal{S}_m} 2^m \Delta_a$$

$$\leq \sum_{m=1}^{M} \sum_{a \in \mathcal{S}_m, \Delta_a \geq \beta\sqrt{n/2^m}} \left( \frac{512n\beta^2}{\Delta_a} + \frac{2D\Delta_a}{3n} + 2\Delta_a \right)$$

$$+ \sum_{m=1}^{M} \sum_{a \in \mathcal{S}_m, \Delta_a \leq \beta\sqrt{n/2^m}} (2^m \Delta_a)$$

$$\leq \mathcal{O}(n\sqrt{|\mathcal{T}_B| \log(KT)} + D\Delta_{\max} \log(T/n)).$$

**Step 2** To obtain the other regret bound $\mathcal{O}(\min\{R_1, R_2\} + (nd^\star + DB) \log(T/n) + D\Delta_{\max})$ with a different delay overhead, we similarly define

$$\mathsf{rad}_{m,a}^N = \beta \sum_{i=1}^{|\mathcal{S}_m|} |\lambda_{m,i}^{(a)}| \cdot \frac{\|a\|_2}{\sqrt{2^m}}$$

as the total confidence radius of action $a$ coming from the definition of $\mathrm{LCB}_{m,1}(a)$. Further let $\widehat{\mu}_m(a) =$

$\frac{1}{2^m} \left( \sum_{\tau \in O_m(a) \cup E_m(a)} u_\tau \right)$ be the empirical mean of action $a$'s loss within epoch $m$ (which is generally not available to the algorithm due to delay). According to the construction of $\widehat{\mu}_m^+(a)$ and $\widehat{\mu}_m^-(a)$, we know that for all $a \in \mathcal{S}_m$,

$$\widehat{\mu}_m^+(a) \leq \widehat{\mu}_m(a) + \frac{|E_m(a)|}{2^m}, \quad \widehat{\mu}_m^-(a) \geq \widehat{\mu}_m(a) - \frac{|E_m(a)|}{2^m}.$$

Then, for any action $a \in \mathcal{A}_m$ that is not eliminated at the end of epoch $m$, using the fact that $a = \sum_{i=1}^{|\mathcal{S}_m|} \lambda_{m,i}^{(a)} a_{m,i}$, we obtain with high probability:

$$\mu_a \leq \sum_{i=1}^{|\mathcal{S}_m|} \lambda_{m,i}^{(a)} \cdot \widehat{\mu}_m(a_{m,i}) + \mathsf{rad}_{m,a}^N$$

$$\leq \mathrm{LCB}_{m,1}(a) + \mathsf{rad}_{m,a}^N + \sum_{i=1}^{|\mathcal{S}_m|} |\lambda_{m,i}^{(a)}| \cdot \frac{|E_m(a_{m,i})|}{2^m}$$

$$\leq \mathrm{LCB}_{m,1}(a) + \mathsf{rad}_{m,a}^N$$
$$+ \sum_{i=1}^{|\mathcal{S}_m|} |\lambda_{m,i}^{(a)}| \cdot \left( \frac{2D\mu_{a_{m,i}}}{2^m |\mathcal{S}_m|} + \frac{16 \log KT + 2}{2^m} \right) \tag{15}$$

$$\leq B + \mathsf{rad}_{m,a}^N$$
$$+ \sum_{i=1}^{|\mathcal{S}_m|} |\lambda_{m,i}^{(a)}| \cdot \left( \frac{2D\mu_{a_{m,i}}}{2^m |\mathcal{S}_m|} + \frac{16 \log KT + 2}{2^m} \right), \tag{16}$$

where the first inequality is by standard Azuma-Hoeffding's inequality, the third inequality is by Lemma C.2 of Schlisselberg et al. (2025) (included as Lemma A.2 in the appendix for completeness), and the last inequality is because $a$ is not eliminated at the end of epoch $m$.

Now consider two cases. When $B \geq \frac{\mu_a}{2}$, we know that $\Delta_a \leq \mu_a - \mu^\star \leq 2B$. Using the previous Eq. (14), we know that

$$2^m \Delta_a \leq \mathcal{O}\left( \frac{n\beta^2}{\Delta_a} + \frac{DB}{n} \right). \tag{17}$$

Otherwise, when $B < \frac{\mu_a}{2}$, with some manipulation on Eq. (16), we show that

$$2^m \Delta_a \leq \mathcal{O}\left( \frac{n\beta^2}{\Delta_a} + \frac{\sum_{i=1}^{|\mathcal{S}_m|} D\mu_{a_{m,i}}}{n} \right). \tag{18}$$

Combining Eq. (17) and Eq. (18), we then obtain that within epoch $m$, the regret is bounded by

$$\mathcal{O}\left( \sum_{a \in \mathcal{S}_m} \frac{n\beta^2}{\Delta_a} + DB + D \sum_{i=1}^{|\mathcal{S}_{m-1}|} \mu_{a_{m-1,i}} \right), \tag{19}$$

since all active actions in epoch $m$ are not eliminated in epoch $m-1$. The first term $\sum_{a \in \mathcal{S}_m} \frac{n\beta^2}{\Delta_a}$ in Eq. (19)

---

**Algorithm 2:** Reduction from Contextual Linear Bandits to Non-Contextual Linear Bandits (Hanna et al., 2023)

---

Input: confidence level $\delta$, an instance $\mathsf{Alg}_{\text{n-ctx}}$ of Algorithm 3 with $\beta = \sqrt{2\log(KT^3)}$.

Let $\Theta'$ be a $\frac{1}{T}$-cover of $\Theta$ with size $\mathcal{O}(T^n)$.

**for** $m = 1, 2, \ldots$ **do**

1    Construct action set $\mathcal{X}_m = \{g^{(m)}(\theta) \mid \theta \in \Theta'\}$ where $g^{(m)}(\theta) = \frac{1}{2^{m-1}} \sum_{\tau=1}^{2^{m-1}} \mathrm{argmin}_{a \in \mathcal{A}_\tau} \langle a, \theta \rangle$.

2    Initiate $\mathsf{Alg}_{\text{n-ctx}}$ with action set $\mathcal{X}_m$ and misspecification level $\varepsilon_m = \min\{1, 2\sqrt{\log(T|\Theta'|/\delta)/2^m}\}$.

3    **for** $t = 2^{m-1} + 1, \ldots, 2^m$ **do**

4      $\mathsf{Alg}_{\text{n-ctx}}$ outputs action $g^{(m)}(\theta_t)$.

5      Observe $\mathcal{A}_t$ and select $a_t = \mathrm{argmin}_{a \in \mathcal{A}_t} \langle a, \theta_t \rangle$.

6      Observe the loss $u_\tau$ for all $\tau$ such that $\tau + d_\tau \in (t-1, t]$ and send them to $\mathsf{Alg}_{\text{n-ctx}}$.

---

eventually leads to the $\min\{R_1, R_2\}$ term in the claimed regret bound, by the exact same reasoning as in **Step 1**. The second term explains the final $DB\log(T/n)$ term in the regret bound (recall that number of epoch is of order $\mathcal{O}(\log(T/n))$). Finally, the last term in Eq. (19) can be written as $D \sum_{i=1}^{|\mathcal{S}_{m-1}|} \Delta_{a_{m-1,i}} + 3n \cdot d^\star$, and the term $D \sum_{i=1}^{|\mathcal{S}_{m-1}|} \Delta_{a_{m-1,i}}$ is one half of the regret incurred in epoch $m-1$ as long as $2^{m-1} > 2D$ (otherwise, the epoch length is smaller than $D$, and we bound the regret trivially by $D\Delta_{\max}$). Summing over all epochs and rearranging the terms thus leads to the a term $nd^\star \log(T/n)$ in the regret. This proves the goal of the second step.

**Combining everything** Finally, note that the number of different values of $B$ Algorithm 1 uses is upper bounded by $\lceil \log_2(d^\star) \rceil = \lceil \log_2(D\mu^\star) \rceil$ since the optimal action $a^\star$ will never be eliminated when $B \geq \mu^\star$. Summing up the regret over these different values of $B$ arrives at the the final bound $\mathcal{O}(\min\{V_1, V_2\}, \log(d^\star)\min\{W_1, W_2\})$.

## 4. Extension to Contextual Linear Bandits

In this section, we extend our results to the stochastic contextual setting where the action set at each round is drawn i.i.d. from a distribution $\mathcal{P}$. While the arm elimination procedure is critical in solving our problem in the non-contextual case with a fixed action set, it is not clear (if possible at all) to directly generalize it to the contextual setting due to the dynamic nature of the action set.

Fortunately, a recent work by Hanna et al. (2023) proposes a reduction from contextual linear bandits to non-contextual linear bandits (both without delay). At a high level, this reduction constructs a fixed action for each possible parameter $\theta$ of the contextual bandit instance, where the expected payoff of this fixed action equals to the expected payoff of the optimal action assuming that the true parameter is $\theta$. Since the distribution of the action set is unknown, this reduction estimates each action's expected payoff using the historical action sets. This estimation introduces a misspecification

error, resulting in a misspecified model. Therefore, the subroutine used by this reduction needs to be able to handle an $\varepsilon$-misspecified model, where the loss of each $a \in \mathcal{A}$ is approximately linear: $\mu_a = \langle a, \theta \rangle + \varepsilon_a \in [0, 1]$, with $\varepsilon \geq \max_{a \in \mathcal{A}} |\varepsilon_a|$ indicating the misspecification level. It turns out that, a simple modification of our Algorithm 1 can indeed address such misspecification — it only requires incorporating the misspecification level $\varepsilon$ into the criteria of arm elimination; see Algorithm 3 and specifically its Line 10 for details.

We then plugin this subroutine, denoted as $\mathsf{Alg}_{\text{n-ctx}}$, into their reduction, as shown in Algorithm 2. Specifically, the algorithm first constructs a $\frac{1}{T}$-cover $\Theta'$ of the parameter space $\Theta = \mathbb{R}_+^n \cap \mathbb{B}_2^n(1)$ with size $|\Theta'| = \mathcal{O}(T^n)$. It then proceeds in epochs with doubling length. At the start of epoch $m$, a new *fixed* action set $\mathcal{X}_m = \{g^{(m)}(\theta) : \theta \in \Theta'\}$ is constructed, where $g^{(m)}(\theta)$ is the averaged optimal action over the previous $m-1$ epochs, assuming the model parameter being $\theta$. Then, a new instance of $\mathsf{Alg}_{\text{n-ctx}}$ with action set $\mathcal{X}_m$ and some misspecification level $\varepsilon_m$ is initiated and run for the entire epoch. At each round $t$ of this epoch, $\mathsf{Alg}_{\text{n-ctx}}$ outputs an action $g^{(m)}(\theta_t) \in \mathcal{X}_m$, and the algorithm's final decision upon receiving the true action set $\mathcal{A}_t$ is $a_t = \mathrm{argmin}_{a \in \mathcal{A}_t} \langle a, \theta_t \rangle$. Finally, at the end of this round, all newly observed losses are sent to $\mathsf{Alg}_{\text{n-ctx}}$.

**Guarantees and Analysis** Even though our algorithm is a direct application of the reduction of Hanna et al. (2023), it is a priori unclear whether it enjoys any favorable regret guarantee in the delay-as-loss setting. By adopting and generalizing their analysis, we show that this is indeed the case. Before introducing our results, we define the following quantities:

$$g(\theta) \triangleq \mathbb{E}_{\mathcal{A} \sim \mathcal{P}} \left[ \mathrm{argmin}_{a \in \mathcal{A}} \langle a, \theta \rangle \right],$$

$$\Delta_{\min}^{\text{n-ctx}} \triangleq \min_{\theta' \in \Theta', \langle g(\theta'), \theta \rangle \neq \langle g(\theta), \theta \rangle} \mathbb{E}\left[ \langle g(\theta) - g(\theta'), \theta \rangle \right],$$

$$\Delta_{\max}^{\text{n-ctx}} \triangleq \max_{\theta' \in \Theta'} \mathbb{E}\left[ \langle g(\theta) - g(\theta'), \theta \rangle \right],$$

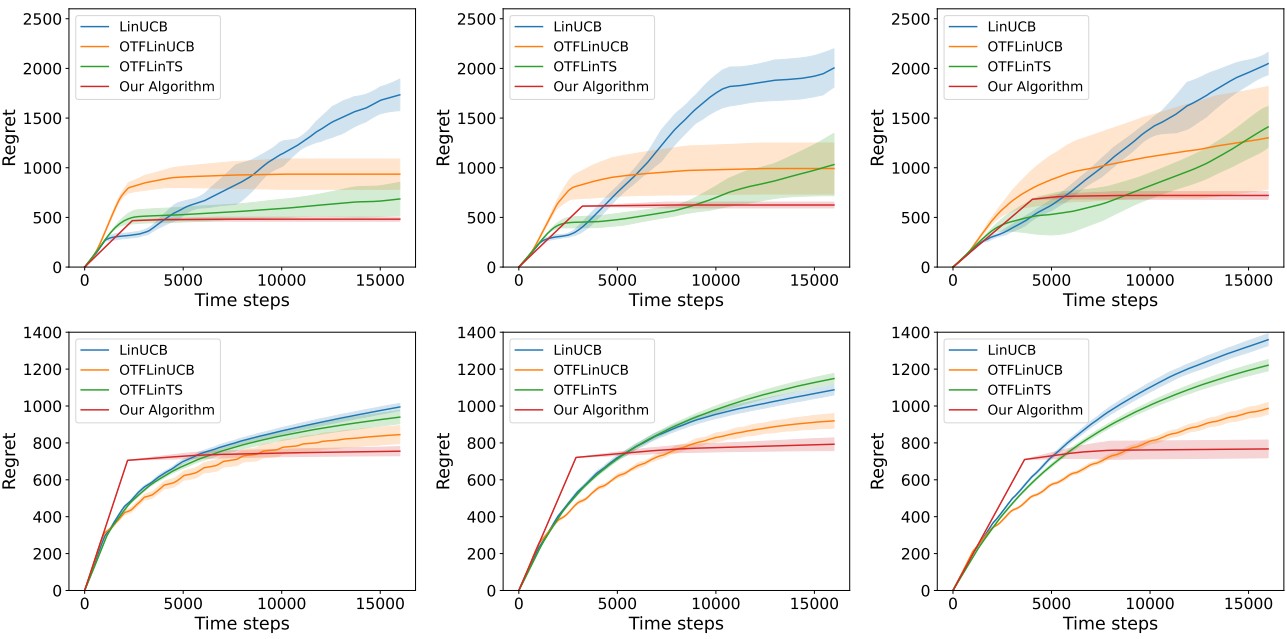

*Figure 1.* Comparison of the empirical results of our algorithm, `LinUCB`, `OTFLinUCB`, and `OTFLinTS` with $K = 70$. The top row is the delay-as-loss setting and the bottom row is the delay-as-reward setting. The left, middle, and right column correspond to $n = 6, 8, 10$ respectively.

$$\overline{d}^{\star} \triangleq D \cdot \langle g(\theta), \theta \rangle = D \cdot \mathbb{E}_{\mathcal{A} \sim \mathcal{P}} \left[ \min_{a \in \mathcal{A}} \langle a, \theta \rangle \right],$$

where $g(\theta)$ denotes the optimal action in expectation, $\Delta_{\min}^{\text{n-ctx}}$ ($\Delta_{\max}^{\text{n-ctx}}$) denotes the minimum (maximum) suboptimality gap for the reduced non-contextual linear bandit instance, and $\overline{d}^{\star}$ denotes the expected delay of the optimal action.

**Theorem 4.1.** *Algorithm 2 with $\delta = 1/T^2$ guarantees*

$$\text{Reg} = \mathcal{O}\big(n\sqrt{T \log T} + \min\{V_1, V_2\}$$
$$+ \log(\overline{d}^{\star}) \min\{W_1, W_2\}\big),$$

*where* $V_1 = \frac{n^3 \log^2(T) \log(T/n) \log(\overline{d}^{\star})}{\Delta_{\min}^{\text{n-ctx}}}$, $V_2 = n^{1.5}\sqrt{T \log(\overline{d}^{\star}) \log(T)}$, $W_1 = \log T(n\overline{d}^{\star} \log(T/n) + D\Delta_{\max}^{\text{n-ctx}})$, *and* $W_2 = D\Delta_{\max}^{\text{n-ctx}} \log T \log(T/n)$.

The proof is deferred to Appendix C. The regret bound is in the same spirit as the one for the non-contextual case (Theorem 3.3) and consists of a term for standard regret and a term for delay overhead. The standard part unfortunately suffers higher dependence on the dimension $n$, while the delay overhead is in a similar problem-dependent form. We remark that this is the first regret guarantee for contextual linear bandits with delay-as-payoff, resolving an open problem asked by (Schlisselberg et al., 2025).

## 5. Experiment

In this section, we implement and evaluate our algorithm for both the delay-as-loss and delay-as-reward settings. For simplicity, we only consider the non-contextual setting. Since there are no existing algorithms for this problem (to the best of our knowledge), we consider three simple benchmarks. The first one applies the standard `LinUCB` algorithm only using the currently available observations (see Algorithm 5 in Appendix D for details). We point out that this simple approach to handling delayed feedback is indeed very common in the literature and in fact enjoys favorable guarantees at least for some problems (Thune et al., 2019; van der Hoeven et al., 2023). Additionally, we include two benchmark algorithms: `OTFLinUCB` and `OTFLinTS` (Vernade et al., 2020a), which are designed for linear bandits but with payoff-independent stochastic delay.

**Experiment setup** The experiment setup is as follows. We set the dimension $n \in \{6, 8, 10\}$ and the size of the action set $|\mathcal{A}| = 70$. The model parameter $\theta$ is set to be $\frac{|\nu|}{\|\nu\|_2}$ where $\nu$ is drawn from the $n$-dimensional standard normal distribution and $|\nu|$ denotes the entry-wise absolute value of $\nu$ to make sure that $\theta \in \mathbb{R}_+^n \cap \mathbb{B}_2^n(1)$. Each action $a \in \mathcal{A}$ is constructed by first sampling $a_i$ uniformly from $[0, 1]$ for all $i \in [n]$ and then normalizing it to unit $\ell_2$-norm. When an action $a_t$ is chosen at round $t$, the payoff $u_t$ is drawn from beta distribution with $\alpha = \mu_{a_t}$ and $\beta = 1 - \mu_{a_t}$. The

number of iterations $T$ is $16000$ and the maximal possible delay $D$ is $1000$. For simplicity, we also ignore the role of $B$ in our algorithms.

**Results**    In Figure 1, we plot the mean and the standard deviation of the regret over $8$ independent experiments with different random seeds, for each $n \in \{6, 8, 10\}$ (the columns) and also both delay-as-loss and delay-as-reward (the rows). We observe that our algorithm consistently outperforms all the three benchmarks in all setups. Also, in all runs, after about 9 to 12 epochs, our algorithm eliminates a significant number of bad actions, leading to almost constant regret after that point (and explaining the "phase transition" in the plots).

## 6. Conclusion

In this work, we initiate the study of the delay-as-payoff model for contextual linear bandits and develop provable algorithms that require novel ideas compared to standard linear bandits. Interesting future directions include proving matching regret lower bounds and extending our results to general payoff-dependent delays (Lancewicki et al., 2021), unbounded delay/payoff (Howson et al., 2023; Zhou et al., 2019), and other even more challenging settings, such as those with intermediate observations (Vernade et al., 2020b; Esposito et al., 2023) or evolving observations (Bar-On & Mansour, 2024).

## Impact Statement

This paper presents work whose goal is to advance the field of Machine Learning. There are many potential societal consequences of our work, none which we feel must be specifically highlighted here.

## Acknowledgement

HL is supported by NSF award IIS-1943607.

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

# A. Omitted Details in Section 3

In this section, we provide the detailed proof for Theorem 3.3. Specifically, as mentioned in Section 4, we prove the guarantee of a modified algorithm (Algorithm 3) for the more general $\varepsilon$-misspecified linear bandits.

Recall that in misspecified linear bandits, $\mu_a = \langle a, \theta \rangle + \varepsilon_a \in [0,1]$ with $|\varepsilon_a| \leq \varepsilon$ for all $a \in \mathcal{A}$. Due to this difference, we clarify on the definitions of $\Delta_a$, $a^\star$, $\mu^\star$, $\Delta_{\min}$, $\Delta_{\max}$, and $d^\star$ in misspecified linear bandits as follows. We still define $\Delta_a = \langle a^\star - a, \theta \rangle$ as the suboptimality gap of action $a$, where $a^\star \in \operatorname{argmin}_{a \in \mathcal{A}} \langle a, \theta \rangle$, but $\mu^\star \triangleq \min_{a \in \mathcal{A}} \mu_a$ as the loss of the optimal action. Note that due to the misspecification, $\mu^\star$ may not necessarily be $\mu_{a^\star}$. Define $\Delta_{\min} = \min_{a \in \mathcal{A}, \Delta_a > 0} \Delta_a$ and $\Delta_{\max} = \max_{a \in \mathcal{A}} \Delta_a$ to be the minimum non-zero, and maximum sub-optimality gap. The delay at round $t$ is still defined as $d_t = D \cdot u_t$ and $d^\star = D \cdot \mu^\star$ is the expected delay of the optimal action.

As for the algorithm, Algorithm 3 differs from Algorithm 1 only in Line 10 where we add one misspecification term $4\sqrt{3n}\varepsilon$ in the criteria of eliminating an action.

The following theorem shows the guarantee of our algorithm in the misspecified linear bandits.

**Theorem A.1.** *Algorithm 3 with $\beta = \sqrt{2\log(KT^3)}$ guarantees that*

$$
\operatorname{Reg} \leq \mathcal{O}\left(\min\left\{\frac{n^2 \log(KT)\log(T/n)\log(d^\star)}{\Delta_{\min}}, n\sqrt{T\log(d^\star)\log(KT)}\right\} + \varepsilon\sqrt{n}T\right)
$$
$$
+ \log(d^\star) \cdot \mathcal{O}\left(\min\left\{nd^\star \log(T/n) + D\Delta_{\max}, D\Delta_{\max}\log(T/n)\right\}\right).
$$

To prove Theorem A.1, recall the following quantities

$$
\widehat{\mu}_m(a) = \frac{1}{2^m} \sum_{\tau \in O_m(a) \cup E_m(a)} u_\tau, \quad \forall a \in \mathcal{S}_m, \tag{29}
$$

$$
\widehat{\mu}_{m,1}(a) = \sum_{i=1}^{|\mathcal{S}_m|} \lambda_{m,i}^{(a)} \cdot \widehat{\mu}_m(a_{m,i}), \quad \forall a \in \mathcal{A}_m, \tag{30}
$$

$$
\widehat{\mu}_{m,2}(a) = \sum_{i=1}^{|\mathcal{S}_m|} \lambda_{m,i}^{(a)} \cdot \widehat{\mu}_m^F(a_{m,i}), \quad \forall a \in \mathcal{A}_m. \tag{31}
$$

We then define the following event and show that the event holds with high probability.

*Event* 1. For all action $a \in \mathcal{A}_m, m \in [T]$,

$$
|\langle a, \theta \rangle - \widehat{\mu}_{m,1}(a)| \leq \sqrt{|\mathcal{S}_m|}\varepsilon + \beta \sum_{i=1}^{|\mathcal{S}_m|} \left|\lambda_{m,i}^{(a)}\right| \sqrt{\frac{1}{2^m}}, \tag{32}
$$

$$
|\langle a, \theta \rangle - \widehat{\mu}_{m,2}(a)| \leq \sqrt{|\mathcal{S}_m|}\varepsilon + \beta \sum_{i=1}^{|\mathcal{S}_m|} \left|\lambda_{m,i}^{(a)}\right| \sqrt{\frac{1}{c_m(a_{m,i})}}, \tag{33}
$$

$$
|E_m(a)| \leq \frac{2D\mu_a}{|\mathcal{S}_m|} + 16\log KT + 2, \tag{34}
$$

where $\beta = \sqrt{2\log KT^3}$.

**Lemma A.2.** *Algorithm 3 guarantees that Event 1 holds with probability at least $1 - \frac{2}{T^2}$.*

*Proof.* Fix an action $a \in \mathcal{S}_m$ in epoch $m \in [T]$. According to standard Azuma's inequality, we know that with probability at least $1 - \delta$,

$$
|\mu_a - \widehat{\mu}_{m,1}(a)| \leq \sqrt{\frac{2\log(2/\delta)}{2^m}} \|a\|_2,
$$

$$
|\mu_a - \widehat{\mu}_{m,2}(a)| \leq \sqrt{\frac{2\log(2/\delta)}{c_m(a)}} \|a\|_2.
$$

---

**Algorithm 3:** Phased Elimination via Volumetric Spanner for Linear Bandits with Delay-as-Loss with misspecification

---

1 Input: maximum possible delay $D$, action set $\mathcal{A}$, $\beta > 0$, a misspecification level $\varepsilon$.

2 Initialization: optimal loss guess $B = 1/D$.

3 Initialization: active action set $\mathcal{A}_1 = \mathcal{A}$.

   **for** $m = 1, 2, \ldots,$ **do**

4      Find $\mathcal{S}_m = \{a_{m,1}, \ldots, a_{m,|\mathcal{S}_m|}\}$, a volumetric spanner of $\mathcal{A}_m$ with $|\mathcal{S}_m| = 3n$.

5      Pick each $a \in \mathcal{S}_m$ $2^m$ times in a round-robin way.

6      Let $\mathcal{I}_m$ contain all the rounds in this epoch.

7      For each $a \in \mathcal{S}_m$, calculate the following quantities:

$$\widehat{\mu}_m^+(a) = \frac{1}{2^m}\Big( \sum_{\tau \in O_m(a)} u_\tau + \sum_{\tau \in E_m(a)} 1 \Big), \tag{20}$$

$$\widehat{\mu}_m^-(a) = \frac{1}{2^m} \sum_{\tau \in O_m(a)} u_\tau, \tag{21}$$

$$\widehat{\mu}_{m,1}^+(a) = \widehat{\mu}_m^+(a) + \frac{\beta}{2^{m/2}}\|a\|_2, \tag{22}$$

$$\widehat{\mu}_{m,1}^-(a) = \widehat{\mu}_m^-(a) - \frac{\beta}{2^{m/2}}\|a\|_2, \tag{23}$$

$$\widehat{\mu}_m^F(a) = \frac{1}{c_m(a)} \sum_{\tau \in C_m(a)} u_\tau, \tag{24}$$

$$\widehat{\mu}_{m,2}^+(a) = \widehat{\mu}_m^F(a) + \frac{\beta}{\sqrt{c_m(a)}}\|a\|_2, \tag{25}$$

$$\widehat{\mu}_{m,2}^-(a) = \widehat{\mu}_m^F(a) - \frac{\beta}{\sqrt{c_m(a)}}\|a\|_2, \tag{26}$$

    where $c_m(a) = |C_m(a)|$, $C_m(a) = \{\tau \in \mathcal{I}_m : \tau + D \in \mathcal{I}_m, a_\tau = a\}$,
    $O_m(a) = \{\tau \in \mathcal{I}_m : \tau + d_\tau \in \mathcal{I}_m, a_\tau = a\}$, and $E_m(a) = \{\tau \in \mathcal{I}_m : a_\tau = a\} \setminus O_m(a)$.

    **for** *each* $a \in \mathcal{A}_m$ **do**

8        Decompose $a$ as $a = \sum_{i=1}^{|S_m|} \lambda_{m,i}^{(a)} a_{m,i}$ with $\|\lambda_m^{(a)}\|_2 \le 1$ and calculate

$$\mathrm{UCB}_m(a) = \sum_{i=1}^{|\mathcal{S}_m|} \lambda_{m,i}^{(a)} \cdot \widehat{\mu}_{m,2}^{\mathrm{SGN}(\lambda_{m,i}^{(a)})}(a_{m,i}), \tag{27}$$

$$\mathrm{LCB}_m(a) = \max_{j \in \{1,2\}} \{\mathrm{LCB}_{m,j}(a)\} \ \text{ where}$$

$$\mathrm{LCB}_{m,j}(a) = \sum_{i=1}^{|\mathcal{S}_m|} \lambda_{m,i}^{(a)} \cdot \widehat{\mu}_{m,j}^{\mathrm{SGN}(-\lambda_{m,i}^{(a)})}(a_{m,i}), \tag{28}$$

9      Set $\mathcal{A}_{m+1} = \mathcal{A}_m$.

     **for** $a \in \mathcal{A}_m$ **do**

10        **if** $\exists a' \in \mathcal{A}_m$, *s.t.* $\mathrm{LCB}_m(a) \ge \min\{\mathrm{UCB}_m(a'), B\} + 4\sqrt{3n}\varepsilon$ **then**
          Eliminate $a$ from $\mathcal{A}_{m+1}$.

11     **if** $\mathcal{A}_{m+1} = \emptyset$ **then**
       Set $B \leftarrow 2B$ and go to Line 3.

---

Taking union bound over all possible $a \in \mathcal{A}$ and all $m \in [T]$, we know that with probability at least $1 - \delta$, for all $a \in \mathcal{S}_m$ and all $m \in [T]$,

$$|\mu_a - \widehat{\mu}_{m,1}(a)| \leq \sqrt{\frac{2\log(2TK/\delta)}{n_t(a)}} \|a\|_2,$$

$$|\mu_a - \widehat{\mu}_{m,2}(a)| \leq \sqrt{\frac{2\log(2TK/\delta)}{c_m(a)}} \|a\|_2.$$

Then, given that the above equation holds, for $a \in \mathcal{A}_m$, due to the property of volumetric spanners, we have $\mu_a = \langle a, \theta^\star \rangle + \varepsilon_a = \sum_{i=1}^{|\mathcal{S}_m|} \lambda_{m,i}^{(a)} \langle a_{m,i}, \theta^\star \rangle + \varepsilon_a$. Therefore, we can obtain that

$$
\begin{aligned}
|\langle a, \theta \rangle - \widehat{\mu}_{m,1}(a)| &\leq \left| \sum_{i=1}^{|\mathcal{S}_m|} \lambda_{m,i}^{(a)} (\langle a_{m,i}, \theta^\star \rangle - \mu_{a_{m,i}}) \right| + \sum_{i=1}^{|\mathcal{S}_m|} \left| \lambda_{m,i}^{(a)} \right| \cdot \left| \mu_{a_{m,i}} - \widehat{\mu}_m(a_{m,i}) \right| \\
&\leq \sum_{i=1}^{|\mathcal{S}_m|} \left| \lambda_{m,i}^{(a)} \right| \left( \varepsilon_{a_{m,i}} + \sqrt{\frac{2\log(2TK/\delta)}{2^m}} \right) \\
&\leq \sqrt{|\mathcal{S}_m|} \varepsilon + \sum_{i=1}^{|\mathcal{S}_m|} \left| \lambda_{m,i}^{(a)} \right| \sqrt{\frac{2\log(2TK/\delta)}{2^m}},
\end{aligned}
$$

where the last inequality uses $\|\lambda_m^{(a)}\|_1 \leq \sqrt{|\mathcal{S}_m|} \cdot \|\lambda_m^{(a)}\|_2 \leq \sqrt{|\mathcal{S}_m|}$. A similar analysis proves Eq. (33). Eq. (34) holds with probability at least $1 - \frac{1}{T^2}$ according to Lemma 4.1 of (Schlisselberg et al., 2025). Picking $\delta = \frac{1}{T^2}$ finishes the proof. $\qquad\square$

The next lemma shows that if $B \geq \mu^\star$, then Algorithm 3 will not reach an empty active set.

**Lemma A.3.** *Suppose that Event 1 holds. If $B \geq \mu^\star$, then $a^\star \in \mathcal{A}_m$ for all $m$.*

*Proof.* Since Event 1 holds, we have, we know that for all $a \in \mathcal{A}_m$, $\mathrm{LCB}_m(a) \leq \langle a, \theta \rangle + \sqrt{|\mathcal{S}_m|}\varepsilon$ and $\mathrm{UCB}_m(a) \geq \langle a, \theta \rangle - \sqrt{|\mathcal{S}_m|}\varepsilon$. If $B \geq \mu^\star$, then we have $a^\star$ never eliminated since for any $a \in \mathcal{A}_m$

$$
\begin{aligned}
\mathrm{LCB}_m(a^\star) &\leq \langle a^\star, \theta \rangle + \varepsilon\sqrt{|\mathcal{S}_m|} \leq \mu^\star + \varepsilon + \varepsilon\sqrt{|\mathcal{S}_m|} \leq \mu^\star + 2\varepsilon\sqrt{|\mathcal{S}_m|}, \\
\mathrm{LCB}_m(a^\star) &\leq \langle a^\star, \theta \rangle + \varepsilon\sqrt{|\mathcal{S}_m|} \leq \langle a, \theta \rangle + 2\varepsilon\sqrt{|\mathcal{S}_m|} \leq \mathrm{UCB}_m(a) + 4\varepsilon\sqrt{|\mathcal{S}_m|}.
\end{aligned}
$$

Therefore, $a^\star$ never satisfy the elimination condition. $\qquad\square$

The following lemma shows that the regret within epoch $m$ can be well-controlled.

**Lemma A.4.** *Suppose that Event 1 holds. Algorithm 3 guarantees that if $a \in \mathcal{A}$ is not eliminated at the end of epoch $m$ (meaning that $a \in \mathcal{A}_{m+1}$), then*

$$2^m \cdot \Delta_a \leq 2^m \cdot 24\sqrt{n}\varepsilon + \frac{256n\beta^2}{\Delta_a} + \frac{2D\Delta_a}{|\mathcal{S}_m|}.$$

*Proof.* For notational convenience, define $\mathrm{rad}_{m,a}^N = \frac{\beta}{\sqrt{2^m}}\|a\|_2$ and $\mathrm{rad}_{m,a}^F = \frac{\beta}{\sqrt{c_m(a)}}\|a\|_2$ for all $a \in \mathcal{S}_m$. In addition, we also define $\mathrm{rad}_{m,a}^N$ and $\mathrm{rad}_{m,a}^F$ for $a \notin \mathcal{S}_m$ as follows:

$$\mathrm{rad}_{m,a}^N = \sum_{i=1}^{|\mathcal{S}_m|} |\lambda_{m,i}^{(a)}| \cdot \mathrm{rad}_{m,a_{m,i}}^N,$$

$$\mathrm{rad}_{m,a}^F = \sum_{i=1}^{|\mathcal{S}_m|} |\lambda_{m,i}^{(a)}| \cdot \mathrm{rad}_{m,a_{m,i}}^F.$$

Since Event 1 holds, we know that for all $a \in \mathcal{A}_m$, $\mathrm{LCB}_m(a) \leq \langle a, \theta \rangle + \sqrt{|\mathcal{S}_m|}\varepsilon$, $\mathrm{UCB}_m(a) \geq \langle a, \theta \rangle - \sqrt{|\mathcal{S}_m|}\varepsilon$. Moreover, as $\mathrm{LCB}_m(a) = \max\{\mathrm{LCB}_{m,1}(a), \mathrm{LCB}_{m,2}(a)\}$, we know that for all $a \in \mathcal{A}_m$

$$\mathrm{LCB}_{m,1}(a) + 2\mathsf{rad}^N_{m,a} + 2\varepsilon\sqrt{|\mathcal{S}_m|} \geq \widehat{\mu}_{m,1}(a) + \mathsf{rad}^N_{m,a} + 2\varepsilon\sqrt{|\mathcal{S}_m|} \geq \langle a, \theta \rangle,$$

$$\mathrm{LCB}_{m,2}(a) + 2\mathsf{rad}^F_{m,a} + 2\varepsilon\sqrt{|\mathcal{S}_m|} \geq \widehat{\mu}_{m,2}(a) + \mathsf{rad}^F_{m,a} + 2\varepsilon\sqrt{|\mathcal{S}_m|} \geq \langle a, \theta \rangle,$$

$$\mathrm{UCB}_m(a) - 2\mathsf{rad}^F_{m,a} - 2\varepsilon\sqrt{|\mathcal{S}_m|} = \widehat{\mu}_{m,2}(a) - \mathsf{rad}^F_{m,a} - 2\varepsilon\sqrt{|\mathcal{S}_m|} \leq \langle a, \theta \rangle.$$

If $B \geq \mu^\star$, then $a^\star \in \mathcal{A}_m$ according to Lemma A.3. Moreover, if $a$ is not eliminated in epoch $m$, we have $\mathrm{LCB}(a) \leq \min\{\mathrm{UCB}_m(a^\star), B\} + 4\sqrt{|S_m|}\varepsilon$, meaning that

$$\langle a, \theta \rangle - 2\mathsf{rad}^F_{m,a} - 2\varepsilon\sqrt{|\mathcal{S}_m|}$$
$$\leq \widehat{\mu}_{m,2}(a) - \mathsf{rad}^F_{m,a}$$
$$\leq \mathrm{LCB}_m(a)$$
$$\leq \min\{\mathrm{UCB}_m(a^\star), B\} + 4\sqrt{|S_m|}\varepsilon$$
$$\leq \mathrm{UCB}_m(a^\star) + 4\sqrt{|S_m|}\varepsilon$$
$$= \widehat{\mu}_{m,2}(a^\star) + \mathsf{rad}^F_{m,a^\star} + 4\sqrt{|S_m|}\varepsilon$$
$$\leq \langle a^\star, \theta \rangle + 2\mathsf{rad}^F_{m,a^\star} + 6\sqrt{|S_m|}\varepsilon.$$

Since $\mathsf{rad}^F_{m,a} = \sum_{i=1}^{|\mathcal{S}_m|} |\lambda^{(a)}_{m,i}| \cdot \mathsf{rad}^F_{m,a_{m,i}}$ with $\|\lambda^{(a)}_m\|_2 \leq 1$, we have that $\|\lambda^{(a)}_m\|_1 \leq \sqrt{|\mathcal{S}_m|}$ and

$$\Delta_a \leq 4\sqrt{|\mathcal{S}_m|}\left(\max_{a \in \mathcal{S}_m} \mathsf{rad}^F_{m,a} + 2\varepsilon\right) = 4\sqrt{3n}\max_{a \in \mathcal{S}_m}\mathsf{rad}^F_{m,a} + 8\sqrt{3n}\varepsilon \leq \frac{8\sqrt{n}\beta}{\min_{a' \in \mathcal{S}_m}\sqrt{c_m(a')}} + 16\sqrt{n}\varepsilon.$$

If $B \leq \mu^\star$, then we have

$$\langle a^\star, \theta \rangle + \varepsilon \geq \mu^\star \geq B \geq \mathrm{LCB}_m(a) - 4\sqrt{|\mathcal{S}_m|}\varepsilon \geq \langle a, \theta \rangle - 2\mathsf{rad}^F_{m,a} - 5\sqrt{|\mathcal{S}_m|}\varepsilon,$$

where the second inequality is because $a$ is not eliminated in epoch $m$. Therefore, we always have

$$\Delta_a \leq 2\mathsf{rad}^F_{m,a} + 6\sqrt{|\mathcal{S}_m|}\varepsilon \leq \frac{8\sqrt{n}\beta}{\min_{a' \in \mathcal{S}_m}\sqrt{c_m(a')}} + 12\sqrt{n}\varepsilon.$$

In addition, we know that for all $a \in \mathcal{S}_m$,

$$2^m \leq c_m(a) + \frac{D}{|\mathcal{S}_m|} + 1 \leq c_m(a) + \frac{2D}{|\mathcal{S}_m|}.$$

Therefore, if $12\sqrt{n}\varepsilon \geq \frac{\Delta_a}{2}$, then we have

$$2^m \Delta_a \leq 2^m \cdot 24\sqrt{n}\varepsilon;$$

otherwise, we have $\Delta_a \leq \frac{8\sqrt{n}\beta}{\min_{a \in \mathcal{S}_m}\sqrt{c_m(a)}} + 12\sqrt{n}\varepsilon \leq \frac{8\sqrt{n}\beta}{\min_{a \in \mathcal{S}_m}\sqrt{c_m(a)}} + \frac{\Delta_a}{2}$ and

$$\Delta_a \leq \frac{16\sqrt{n}\beta}{\min_{a' \in \mathcal{S}_m}\sqrt{c_m(a')}},$$

and we can obtain that

$$\min_{a' \in \mathcal{S}_m} c_m(a') \cdot \Delta_a \leq \frac{256d\beta^2}{\Delta_a}.$$

Combining the above two cases, we know that for all $a \in \mathcal{A}_m$,

$$2^m \cdot \Delta_a \leq 2^m \cdot 24\sqrt{n}\varepsilon + \min_{a' \in \mathcal{S}_m} c_m(a') \cdot \Delta_a + \frac{2D\Delta_a}{|\mathcal{S}_m|} \leq 2^m \cdot 24\sqrt{n}\varepsilon + \frac{256n\beta^2}{\Delta_a} + \frac{2D\Delta_a}{|\mathcal{S}_m|}.$$

$\square$

In fact, the bound above can be obtained by only using $\text{LCB}_{m,1}$. Next, we provide yet-another regret bound within epoch $m$, which utilizes $\text{LCB}_{m,2}$.

**Lemma A.5.** *Algorithm 3 guarantees that under Event 1, if action $a$ is not eliminated at the end of epoch $m$ (meaning that $a \in \mathcal{A}_{m+1}$), then*

$$\langle a, \theta \rangle \leq B + \text{rad}_{m,a}^N + \sum_{i=1}^{|\mathcal{S}_m|} |\lambda_{m,i}^{(a)}| \cdot \left( \frac{2D\mu_{a_{m,i}}}{2^m |\mathcal{S}_m|} + \frac{16 \log T + 2}{2^m} \right) + 8\sqrt{|\mathcal{S}_m|}\varepsilon.$$

*Proof.* For all $a \in \mathcal{S}_m$, since $u_t \in [0, 1]$, we know that

$$\widehat{\mu}_m^+(a) = \frac{1}{2^m} \left( \sum_{\tau \in O_m(a)} u_\tau + \sum_{\tau \in E_m(a)} 1 \right) \leq \widehat{\mu}_{m,a} + \frac{|E_m(a)|}{2^m}, \tag{35}$$

$$\widehat{\mu}_m^-(a) = \frac{1}{2^m} \left( \sum_{\tau \in O_m(a)} u_\tau \right) \geq \widehat{\mu}_{m,a} - \frac{|E_m(a)|}{2^m}. \tag{36}$$

Then, under Event 1, we know that for all $a \in \mathcal{A}_m$,

$$\langle a, \theta \rangle = \sum_{i=1}^{|\mathcal{S}_m|} \lambda_{m,i}^{(a)} \langle a_{m,i}, \theta^\star \rangle$$

$$= \sum_{i=1}^{|\mathcal{S}_m|} \lambda_{m,i}^{(a)} (\mu_{a_{m,i}} - \varepsilon_{a_{m,i}}) \qquad \text{(since } \mu_a = \langle a, \theta^\star \rangle + \varepsilon_a)$$

$$\leq \sum_{i=1}^{|\mathcal{S}_m|} \lambda_{m,i}^{(a)} \cdot \mu_{a_{m,i}} + \sqrt{|\mathcal{S}_m|}\varepsilon \qquad \text{(since } \|\lambda_m^{(a)}\|_1 \leq \sqrt{|\mathcal{S}_m|})$$

$$\leq \sum_{i=1}^{|\mathcal{S}_m|} \lambda_{m,i}^{(a)} \cdot \widehat{\mu}_m(a_{m,i}) + \text{rad}_{m,a}^N + 3\sqrt{|\mathcal{S}_m|}\varepsilon \qquad \text{(since Event 1 holds)}$$

$$\leq \sum_{i=1}^{|\mathcal{S}_m|} \lambda_{m,i}^{(a)} \cdot \widehat{\mu}_m^{sgn(-\lambda_{m,i}^{(a)})}(a_{m,i}) + \text{rad}_{m,a}^N + \sum_{i=1}^{|\mathcal{S}_m|} |\lambda_{m,i}^{(a)}| \cdot \frac{|E_m(a_{m,i})|}{2^m} + 3\sqrt{|\mathcal{S}_m|}\varepsilon \quad \text{(using Eq. (35) and Eq. (36))}$$

$$= \text{LCB}_{m,1}(a) + \text{rad}_{m,a}^N + \sum_{i=1}^{|\mathcal{S}_m|} |\lambda_{m,i}^{(a)}| \cdot \frac{|E_m(a_{m,i})|}{2^m} + 3\sqrt{|\mathcal{S}_m|}\varepsilon$$

$$\leq \text{LCB}_{m,1}(a) + \text{rad}_{m,a}^N + \sum_{i=1}^{|\mathcal{S}_m|} |\lambda_{m,i}^{(a)}| \cdot \left( \frac{2D\mu_{a_{m,i}}}{2^m |\mathcal{S}_m|} + \frac{16 \log KT + 2}{2^m} \right) + 3\sqrt{|\mathcal{S}_m|}\varepsilon. \qquad \text{(since Event 1 holds)}$$

Since $\text{LCB}_{m,1}(a) \leq B + 4\sqrt{|\mathcal{S}_m|}\varepsilon$ (as $a$ is not eliminated at the end of epoch $m$), we have

$$\langle a, \theta \rangle \leq B + \text{rad}_{m,a}^N + \sum_{i=1}^{|\mathcal{S}_m|} |\lambda_{m,i}^{(a)}| \cdot \left( \frac{2D\mu_{a_{m,i}}}{2^m |\mathcal{S}_m|} + \frac{16 \log T + 2}{2^m} \right) + 8\sqrt{|\mathcal{S}_m|}\varepsilon.$$

$\square$

**Lemma A.6.** *If Event 1 holds, Algorithm 3 guarantees that if $a$ is not eliminated at the end of epoch $m$, then we also have*

$$2^m \Delta_a \leq \frac{256n\beta^2}{\Delta_a} + \frac{4DB + 12\sum_{i=1}^{|\mathcal{S}_m|} |\lambda_{m,i}^{(a)}| \cdot D\mu_{a_{m,i}}}{|\mathcal{S}_m|} + (128 \log T + 16)\sqrt{n} + 2^m \cdot 64\sqrt{n}\varepsilon.$$

*Proof.* If $\langle a, \theta \rangle \leq 2B$, we know that $\Delta_a = \langle a - a^\star, \theta \rangle \leq 2B$. Using Lemma A.4, we can obtain that

$$2^m \cdot \Delta_a \leq 2^m \cdot 24\sqrt{n}\varepsilon + \frac{256n\beta^2}{\Delta_a} + \frac{2D\Delta_a}{|\mathcal{S}_m|}$$

$$\leq 2^m \cdot 24\sqrt{n}\varepsilon + \frac{256n\beta^2}{\Delta_a} + \frac{4DB}{|\mathcal{S}_m|}$$

If $\langle a, \theta \rangle \geq 2B$, we have $B \leq \frac{\langle a, \theta \rangle}{2}$. Using Lemma A.5, we know that

$$\Delta_a \leq \langle a, \theta \rangle \leq \underbrace{2 \cdot \mathsf{rad}_{m,a}^N}_{\texttt{Term (1)}} + \underbrace{2\sum_{i=1}^{|\mathcal{S}_m|} |\lambda_{m,i}^{(a)}| \cdot \left( \frac{2D\mu_{a_{m,i}}}{2^m|\mathcal{S}_m|} + \frac{16\log T + 2}{2^m} \right) + 16\sqrt{|\mathcal{S}_m|}\varepsilon}_{\texttt{Term (2)}} \, .$$

If $\texttt{Term (1)} \geq \texttt{Term (2)}$, we have

$$\Delta_a \leq 4\mathsf{rad}_{m,a}^N \varepsilon \leq 4\sqrt{|\mathcal{S}_m|} \max_{a_m \in \mathcal{S}_m} \mathsf{rad}_{m,a_m}^N \leq \frac{8\beta\sqrt{n}}{2^{m/2}},$$

meaning that $2^m \Delta_a \leq \frac{64n\beta^2}{\Delta_a}$. Otherwise, we have

$$\Delta_a \leq 4\sum_{i=1}^{|\mathcal{S}_m|} |\lambda_{m,i}^{(a)}| \cdot \left( \frac{2D\mu_{a_{m,i}}}{2^m|\mathcal{S}_m|} + \frac{16\log T + 2}{2^m} \right) + 64\sqrt{n}\varepsilon,$$

meaning that

$$2^m \Delta_a \leq \frac{8\sum_{i=1}^{|\mathcal{S}_m|} |\lambda_{m,i}^{(a)}| \cdot D\mu_{a_{m,i}}}{|\mathcal{S}_m|} + (128\log T + 16)\sqrt{n} + 2^m \cdot 64\sqrt{n}\varepsilon.$$

Combining both cases, we know that

$$2^m \Delta_a \leq \frac{256n\beta^2}{\Delta_a} + \frac{4DB + 12\sum_{i=1}^{|\mathcal{S}_m|} |\lambda_{m,i}^{(a)}| \cdot D\mu_{a_{m,i}}}{|\mathcal{S}_m|} + (128\log T + 16)\sqrt{n} + 2^m \cdot 64\sqrt{n}\varepsilon.$$

□

Now we are ready to prove our main result Theorem A.1.

*Proof of Theorem A.1.* We analyze the regret when Event 1 holds, which happens with probability at least $1 - \frac{2}{T^2}$. When Event 1 does not hold, the expected regret is bounded by $\frac{2}{T}$.

We then bound the regret with a fixed choice of $B$. Combining Lemma A.4 and Lemma A.5, if action $a$ is not eliminated at the end of epoch $m$, we have

$$2^m \cdot \Delta_a \leq \frac{256n\beta^2}{\Delta_a} + \frac{4DB + 12\sum_{i=1}^{|\mathcal{S}_m|} |\lambda_{m,i}^{(a)}| \cdot D\mu_{a_{m,i}}}{|\mathcal{S}_m|} + (128\log T + 16)\sqrt{n} + 2^m \cdot 64\sqrt{n}\varepsilon,$$

$$2^m \cdot \Delta_a \leq 2^m \cdot 24\sqrt{n}\varepsilon + \frac{256n\beta^2}{\Delta_a} + \frac{2D\Delta_a}{|\mathcal{S}_m|}.$$

Therefore, we have

$$\Delta_a \leq \mathcal{O}\left( \frac{n\beta^2}{2^m \cdot \Delta_a} + \sqrt{n}\varepsilon + \frac{\sqrt{n}\log T}{2^m} \right) + \frac{1}{2^m} \min\left\{ \frac{4DB + 12\sum_{i=1}^{|\mathcal{S}_m|} |\lambda_{m,i}^{(a)}| \cdot D\mu_{a_{m,i}}}{n}, \frac{D\Delta_a}{n} \right\} \, .$$

Denote $\mathcal{T}_B$ to be the number of rounds [Algorithm 3] proceeds with $B$ and define $\mathrm{Reg}_B$ be the expected regret within $\mathcal{T}_B$ rounds. Then, for any $\alpha_m \geq 0$, the overall regret is then upper bounded as follows:

$$
\begin{aligned}
\mathrm{Reg}_B &\triangleq \sum_{m=1}^{\lceil \log_2(|\mathcal{T}_B|/3n\rceil} \sum_{a \in \mathcal{S}_m} 2^m \cdot \Delta_a \\
&\leq \sum_{m=1}^{\lceil \log_2(|\mathcal{T}_B|/3n\rceil} \sum_{a \in \mathcal{S}_m} \mathbb{1}\{\Delta_a > \alpha_m\} \left( \mathcal{O}\left( \frac{n\beta^2}{\Delta_a} + 2^m \sqrt{n}\varepsilon + \sqrt{n}\log T \right) \right. \\
&\qquad + \left. \min\left\{ \frac{4DB + 12\sum_{i=1}^{|\mathcal{S}_{m-1}|}|\lambda_{m-1,i}^{(a)}| \cdot d(a_{m-1,i})}{n}, \frac{2D\Delta_a}{n} \right\} \right) \\
&\qquad\qquad\qquad\qquad\qquad \text{(since $a$ is not eliminated in epoch $m-1$ for all $a \in \mathcal{S}_m$)} \\
&\qquad + \sum_{m \geq 1} \sum_{a \in \mathcal{S}_m} \mathbb{1}\{\Delta_a \leq \alpha_m\} 2^m \Delta_a.
\end{aligned}
$$

Picking $\alpha_m = \beta\sqrt{\frac{n}{2^m}}$, we can obtain that

$$
\begin{aligned}
\mathrm{Reg}_B &= \sum_{m=1}^{\lceil \log_2(|\mathcal{T}_B|/3n\rceil} \sum_{a \in \mathcal{S}_m} \left( \mathcal{O}\left( \beta\sqrt{n \cdot 2^m} + 2^m\sqrt{n}\varepsilon + \sqrt{n}\log T \right) \right. \\
&\qquad + \left. \min\left\{ \frac{4DB + 12\sum_{i=1}^{|\mathcal{S}_{m-1}|}|\lambda_{m-1,i}^{(a)}| \cdot d(a_{m-1,i})}{n}, \frac{2D\Delta_a}{n} \right\} \right) \\
&\leq \mathcal{O}\left( |\mathcal{T}_B|\sqrt{n}\varepsilon + \beta n\sqrt{|\mathcal{T}_B|} + \sqrt{n}\log T \log(T/n) \right) \\
&\qquad + \sum_{m=1}^{\lceil \log_2(|\mathcal{T}_B|/3n)\rceil} \sum_{a \in \mathcal{S}_m} \min\left\{ \frac{4DB + 12\sum_{i=1}^{|\mathcal{S}_{m-1}|}|\lambda_{m-1,i}^{(a)}| \cdot d(a_{m-1,i})}{n}, \frac{2D\Delta_a}{n} \right\}.
\end{aligned}
$$

On the other hand, picking $\alpha_m = 0$, we have

$$
\begin{aligned}
\mathrm{Reg}_B &\leq \sum_{m=1}^{\lceil \log_2(|\mathcal{T}_B|/3n\rceil} \sum_{a \in \mathcal{S}_m} \left( \mathcal{O}\left( \frac{n\beta^2}{\Delta_{\min}} + 2^m\sqrt{n}\varepsilon + \sqrt{n}\log T \right) \right. \\
&\qquad + \left. \min\left\{ \frac{4DB + 12\sum_{i=1}^{|\mathcal{S}_{m-1}|}|\lambda_{m-1,i}^{(a)}| \cdot d(a_{m-1,i})}{n}, \frac{2D\Delta_a}{n} \right\} \right) \\
&\leq \mathcal{O}\left( \frac{n^2\beta^2\log(T/n)}{\Delta_{\min}} + \varepsilon\sqrt{n}|\mathcal{T}_B| + \sqrt{n}\log T \log(T/n) \right) \\
&\qquad + \sum_{m=1}^{\lceil \log_2(|\mathcal{T}_B|/3n\rceil} \sum_{a \in \mathcal{S}_m} \min\left\{ \frac{4DB + 12\sum_{i=1}^{|\mathcal{S}_{m-1}|}|\lambda_{m-1,i}^{(a)}| \cdot d(a_{m-1,i})}{n}, \frac{2D\Delta_a}{n} \right\}.
\end{aligned}
$$

Using the fact that $\beta = \sqrt{2\log(KT^3)}$ and combining both bounds, we can obtain that

$$
\begin{aligned}
\mathrm{Reg}_B &\leq \mathcal{O}\left( \min\left\{ \frac{n^2\log(KT)\log(T/n)}{\Delta_{\min}}, n\sqrt{|\mathcal{T}_B|\log(KT)} \right\} + \varepsilon\sqrt{n}|\mathcal{T}_B| \right) \\
&\qquad + \sum_{m=1}^{\lceil \log_2(|\mathcal{T}_B|/3n\rceil} \sum_{a \in \mathcal{S}_m} \min\left\{ \frac{4DB + 12\sum_{i=1}^{|\mathcal{S}_{m-1}|}|\lambda_{m-1,i}^{(a)}| \cdot d(a_{m-1,i})}{n}, \frac{2D\Delta_a}{n} \right\} \qquad (37)
\end{aligned}
$$

For notational convenience, let $R_B = \mathcal{O}\left( \min\left\{ \frac{n^2\log(KT)\log(T/n)}{\Delta_{\min}}, n\sqrt{|\mathcal{T}_B|\log(KT)} \right\} + \varepsilon\sqrt{n}|\mathcal{T}_B| \right)$. To further analyze this bound, we first upper bound $\min\left\{ \frac{4DB + 12\sum_{i=1}^{|\mathcal{S}_{m-1}|}|\lambda_{m-1,i}^{(a)}| \cdot d(a_{m-1,i})}{n}, \frac{2D\Delta_a}{n} \right\}$ by $\frac{2D\Delta_a}{n}$ and obtain that

$$
\mathrm{Reg}_B \leq R_B + \mathcal{O}\left( D\Delta_{\max}\log(T/n) \right). \qquad (38)
$$

On the other hand, we can also upper bound $\min\left\{\frac{4DB+12\sum_{i=1}^{|\mathcal{S}_{m-1}|}|\lambda_{m-1,i}^{(a)}|\cdot d(a_{m-1,i})}{n}, \frac{2D\Delta_a}{n}\right\}$ by $\frac{12\sum_{i=1}^{|\mathcal{S}_{m-1}|}|\lambda_{m-1,i}^{(a)}|\cdot d(a_{m-1,i})}{n}$ and obtain that

$$\text{Reg}_B \le R_B + \left(\sum_{m=1}^{\lceil\log_2(|\mathcal{T}_B|/3n)\rceil} \sum_{a\in\mathcal{S}_m} \frac{4DB + 12\sum_{i=1}^{|\mathcal{S}_{m-1}|}|\lambda_{m-1,i}^{(a)}|\cdot d(a_{m-1,i})}{n}\right).$$

Let $L_{\mathsf{Alg}}^m = \sum_{a\in\mathcal{S}_m} 2^m\mu_a$ be the total expected loss within epoch $m$ and $L_\star^m = |\mathcal{S}_m|\cdot 2^m\cdot\mu^\star$ be the total expected loss for the optimal action. Define $\text{Reg}_m = L_{\mathsf{Alg}}^m - L_\star^m$. Direct calculation shows that

$$\sum_{a\in\mathcal{S}_m} \frac{\sum_{i=1}^{|\mathcal{S}_{m-1}|}|\lambda_{m-1,i}^{(a)}|\cdot d(a_{m-1,i})}{n}$$

$$\le \frac{3D}{2^{m-1}}\cdot 2^{m-1}\sum_{i=1}^{|\mathcal{S}_{m-1}|}\mu_{a_{m-1,i}} \qquad\qquad (\text{since } |\lambda_{m-1,i}^{(a)}|\le 1 \text{ and } |\mathcal{S}_m| = 3n)$$

$$= \frac{3D}{2^{m-1}}L_{\mathsf{Alg}}^{m-1}.$$

Using the fact that $\text{Reg}_B = \sum_{m=1}^{\lceil\log(|\mathcal{T}_B|/3n)\rceil}\text{Reg}_m$, we know that

$$\sum_{m=1}^{\lceil\log(|\mathcal{T}_B|/3n)\rceil} (L_{\mathsf{Alg}}^m - L_\star^m)$$

$$\le \sum_{m=1}^{\lceil\log(|\mathcal{T}_B|/3n)\rceil} \text{Reg}_m + 2\varepsilon\cdot|\mathcal{T}_B|$$

$$\le R_B + \sum_{m=\lceil\log_2(72D)\rceil+1}^{\lceil\log(|\mathcal{T}_B|/3n)\rceil} \frac{36D}{2^{m-1}}\cdot L_{\mathsf{Alg}}^{m-1} + \sum_{m=1}^{\lceil\log_2(72D)\rceil} 2^m\Delta_{\max} + 12DB\log(T/n) \qquad (2\varepsilon\cdot|\mathcal{T}_B| \text{ is subsumed in } R_B)$$

$$\le R_B + \sum_{m=\lceil\log_2(72D)\rceil+1}^{\lceil\log(|\mathcal{T}_B|/3n)\rceil} \frac{36D}{2^{m-1}}\cdot\left(L_{\mathsf{Alg}}^{m-1} - L_\star^{m-1}\right)$$

$$+ \sum_{m=\lceil\log_2(72D)\rceil+1}^{\lceil\log(|\mathcal{T}_B|/3n)\rceil} \frac{36D}{2^{m-1}}\cdot L_\star^{m-1} + \sum_{m=1}^{\lceil\log_2(72D)\rceil} 2^m\Delta_{\max} + 12DB\log(T/n)$$

$$\le R_B + \frac{1}{2}\sum_{m=\lceil\log_2(72D)\rceil+1}^{\lceil\log(|\mathcal{T}_B|/3n)\rceil}\left(L_{\mathsf{Alg}}^{m-1} - L_\star^{m-1}\right) + 36nD\mu^\star\log(T/(216nD)) + 144D\Delta_{\max} + 12DB\log(T/n)$$

$$= R_B + \frac{1}{2}\sum_{m=\lceil\log_2(72D)\rceil+1}^{\lceil\log(|\mathcal{T}_B|/3n)\rceil}\left(L_{\mathsf{Alg}}^{m-1} - L_\star^{m-1}\right) + 36nd^\star\log(T/(216nD)) + 144D\Delta_{\max} + 12DB\log(T/n).$$

Rearranging the terms, we can obtain that

$$\text{Reg}_B \le R_B + 72nd^\star\log(T/(216nD)) + 288D\Delta_{\max} + 12DB\log(T/n). \tag{39}$$

Combining Eq. (38) and Eq. (39), we know that

$$\text{Reg}_B \le \mathcal{O}\left(\min\left\{\frac{n^2\log(KT)\log(T/n)}{\Delta_{\min}}, n\sqrt{|\mathcal{T}_B|\log(KT)}\right\} + \varepsilon\sqrt{n}|\mathcal{T}_B|\right)$$

$$+ \mathcal{O}\left(\min\left\{nd^\star\log(T/nD) + D\Delta_{\max} + DB\log(T/n), D\Delta_{\max}\log(T/n)\right\}\right). \tag{40}$$

Finally, according to Lemma A.3, Algorithm 3 fails at most $\lceil \log_2(D\mu^\star)) \rceil = \lceil \log_2(d^\star)) \rceil$ times. Summing up the regret over all rounds, we know that the overall regret is bounded as follows:

$$\text{Reg} \leq \sum_{r=0}^{\lceil \log_2(d^\star)) \rceil} \text{Reg}_{2^r/D} \leq \mathcal{O}\left( \min\left\{ \frac{n^2 \log(KT) \log(T/n) \log(d^\star)}{\Delta_{\min}}, n\sqrt{T \log(d^\star) \log(KT)} \right\} + \varepsilon\sqrt{n}T \right)$$
$$+ \log(d^\star) \cdot \mathcal{O}\left( \min\left\{ nd^\star \log(T/n) + D\Delta_{\max}, D\Delta_{\max} \log(T/n) \right\} \right).$$

which finishes the proof. $\square$

# B. Omitted Details for Delay-as-Reward

In this section, we show our results for the delay-as-reward setting. The difference compared with the delay-as-loss setting is that now, $\mu_a = \langle a, \theta \rangle + \varepsilon_a \in [0, 1]$ represents the expected reward of picking action $a$, where $|\varepsilon_a| \leq \varepsilon$ for all $a \in \mathcal{A}$. The learner's goal is to minimize the pseudo regret defined as follows:

$$\text{Reg} \triangleq T \max_{a \in \mathcal{A}} \langle a, \theta \rangle - \mathbb{E}\left[\sum_{t=1}^{T} \langle a_t, \theta \rangle\right]. \tag{41}$$

Define $\Delta_a = \langle a^\star - a, \theta \rangle$ as the suboptimality gap of action $a$, where $a^\star \in \text{argmax}_{a \in \mathcal{A}} \langle a, \theta \rangle$, and $\mu^\star \triangleq \max_{a \in \mathcal{A}} \mu_a$ as the reward of the optimal action. Again, note that due to the misspecification, $\mu^\star$ may not necessarily be $\mu_{a^\star}$. Define $\Delta_{\min} = \min_{a \in \mathcal{A}, \Delta_a > 0} \Delta_a$ to be the minimum non-zero sub-optimality gap. The delay at round $t$ is still defined as $d_t = D \cdot u_t$, and $d^\star = D \cdot \mu^\star$ is the expected delay of the optimal action.

## B.1. Algorithm for Linear Bandits with Delay-as-Reward

We list our algorithm for the reward case in Algorithm 4 for completeness. The algorithm shares the same idea as Algorithm 3.

---

**Algorithm 4:** Phased Elimination for Linear Bandits with Delay-as-Reward

---

1   Input: maximum possible delay $D$, action set $\mathcal{A}$, $\beta > 0$, a misspecification level $\varepsilon$.

2   Initialize optimal reward guess $B = 1$.

3   Initialize active action set $\mathcal{A}_1 = \mathcal{A}$.

4   **for** $m = 1, 2, \ldots,$ **do**

5      Find $\mathcal{S}_m = \{a_{m,1}, \ldots, a_{m,|\mathcal{S}_m|}\}$ to be the volumetric spanner of $\mathcal{A}_m$, where $|\mathcal{S}_m| = 3n$.

6      Pick each $a \in \mathcal{S}_m$ $2^m$ times in a round-robin way.

7      Let $\mathcal{I}_m$ contain all the rounds in this epoch.

8      For all $a \in \mathcal{S}_m$, calculate the following quantities

$$\widehat{\mu}_m^+(a) = \frac{1}{2^m}\Big( \sum_{\tau \in O_m(a)} u_\tau + \sum_{\tau \in E_m(a)} 1 \Big), \tag{42}$$

$$\widehat{\mu}_m^-(a) = \frac{1}{2^m} \sum_{\tau \in O_m(a)} u_\tau, \tag{43}$$

$$\widehat{\mu}_{m,1}^+(a) = \widehat{\mu}_m^+(a) + \frac{\beta}{2^{m/2}}\|a\|_2, \tag{44}$$

$$\widehat{\mu}_{m,1}^-(a) = \widehat{\mu}_m^-(a) - \frac{\beta}{2^{m/2}}\|a\|_2, \tag{45}$$

$$\widehat{\mu}_m^F(a) = \frac{1}{c_m(a)} \sum_{\tau \in C_m(a)} u_\tau, \tag{46}$$

$$\widehat{\mu}_{m,2}^+(a) = \widehat{\mu}_m^F(a) + \frac{\beta}{\sqrt{c_m(a)}}\|a\|_2, \tag{47}$$

$$\widehat{\mu}_{m,2}^-(a) = \widehat{\mu}_m^F(a) - \frac{\beta}{\sqrt{c_m(a)}}\|a\|_2, \tag{48}$$

where $c_m(a) = |C_m(a)|$, $C_m(a) = \{\tau \in \mathcal{I}_m : \tau + D \in \mathcal{I}_m, a_\tau = a\}$, $O_m(a) = \{\tau \in \mathcal{I}_m : \tau + d_\tau \in \mathcal{I}_m, a_\tau = a\}$, and $E_m(a) = \{\tau \in \mathcal{I}_m : a_\tau = a\} \setminus O_m(a)$.

9      **for** *each* $a \in \mathcal{A}_m$ **do**

10         Decompose $a$ as $a = \sum_{i=1}^{|\mathcal{S}_m|} \lambda_{m,i}^{(a)} a_{m,i}$ with $\|\lambda_m^{(a)}\|_2 \leq 1$ and calculate

$$\mathrm{LCB}_m(a) = \sum_{i=1}^{|\mathcal{S}_m|} \lambda_{m,i}^{(a)} \cdot \widehat{\mu}_{m,2}^{-\mathrm{SGN}(\lambda_{m,i}^{(a)})}(a_{m,i}), \tag{49}$$

$$\mathrm{UCB}_m(a) = \max_{j \in \{1,2\}} \{\mathrm{UCB}_{m,j}(a)\} \text{ where}$$

$$\mathrm{UCB}_{m,j}(a) = \sum_{i=1}^{|\mathcal{S}_m|} \lambda_{m,i}^{(a)} \cdot \widehat{\mu}_{m,j}^{\mathrm{SGN}(\lambda_{m,i}^{(a)})}(a_{m,i}), \tag{50}$$

11      Set $\mathcal{A}_{m+1} = \mathcal{A}_m$.

12      **for** $a_1 \in \mathcal{A}_m$ **do**

13         **if** $\exists a_2 \in \mathcal{A}_m,$ *such that* $\max\{\mathrm{LCB}_m(a_2), B\} \geq \mathrm{UCB}_m(a_1) + 4\sqrt{3n}\varepsilon$ **then**

14            Eliminate $a_1$ from $\mathcal{A}_{m+1}$.

15      **if** $\mathcal{A}_{m+1}$ *is empty* **then**

        Set $B \leftarrow B/2$ and go to Line 3.

---

## B.2. Regret Guarantees

In this section, we show the theoretical guarantees for our algorithm in the delay-as-reward setting.

**Theorem B.1.** *[Algorithm 4](#) with $\beta = \sqrt{2\log(KT^3)}$ guarantees that*

$$\mathrm{Reg} \leq \mathcal{O}\left(\min\left\{\frac{n^2\log(KT)\log(T/n)\log(1/\mu^\star)}{\Delta_{\min}}, n\sqrt{T\log(KT)\log(1/\mu^\star)}\right\} + \varepsilon\sqrt{n}T\right)$$

$$+ \mathcal{O}\left(\min\left\{\sum_{j=0}^{\lceil\log_2(1/\mu^\star)\rceil}\sum_{i=1}^{3n} d(a_{m-1,i}^{(2^{-j})}), D\Delta_{\max}\log(1/\mu^\star)\log(T/n)\right\}\right),$$

*where $\{a_{m,i}^B\}_{i=1}^{3n}$ represents the set of volumetric spanner at epoch $m$ with the optimal reward guess $B$.*

Similar to the analysis in [Appendix A](#), our analysis is based on the condition that [Event 1](#) holds, which happens with probability $1 - \frac{2}{T^2}$ according to [Lemma A.2](#). The following lemma is a counterpart of [Lemma A.3](#), providing an upper bound of the number of guesses on the optimal reward $B$.

**Lemma B.2.** *Suppose that [Event 1](#) holds. If $B \leq \mu^\star$, then $a^\star \in \mathcal{A}_m$ for all $m$.*

*Proof.* Since [Event 1](#) holds, we have, we know that for all $a \in \mathcal{A}_m$, $\mathrm{UCB}_m(a) + \sqrt{|\mathcal{S}_m|}\varepsilon \geq \langle a, \theta\rangle$, $\mathrm{LCB}_m(a) + \sqrt{|\mathcal{S}_m|}\varepsilon \leq \langle a, \theta\rangle$ If $B \leq \mu^\star$, then we have $a^\star$ never eliminated since for any $a \in \mathcal{A}_m$,

$$\mathrm{UCB}_m(a^\star) + 2\varepsilon\sqrt{|\mathcal{S}_m|} \geq \max_{a\in\mathcal{A}}\{\langle a, \theta\rangle + \varepsilon_a\} \geq \mu^\star \geq B,$$

$$\mathrm{UCB}_m(a^\star) + 4\varepsilon\sqrt{|\mathcal{S}_m|} \geq \mu^\star + 2\varepsilon\sqrt{|\mathcal{S}_m|} \geq \langle a, \theta\rangle + \varepsilon\sqrt{|\mathcal{S}_m|} \geq \mathrm{LCB}_m(a).$$

Therefore, $a^\star$ never satisfy the elimination condition.

$\square$

The following lemma is a counterpart of [Lemma A.4](#).

**Lemma B.3.** *Suppose that [Event 1](#) holds. [Algorithm 4](#) guarantees that if $a \in \mathcal{A}$ is not eliminated at the end of epoch $m$ (meaning that $a \in \mathcal{A}_{m+1}$), then*

$$2^m \cdot \Delta_a \leq 2^m \cdot 24\sqrt{n}\varepsilon + \frac{256n\beta^2}{\Delta_a} + \frac{2D\Delta_a}{|\mathcal{S}_m|}.$$

*Proof.* Since [Event 1](#) holds, we know that for all $a \in \mathcal{A}_m$, $\mathrm{LCB}_m(a) \leq \mu_a + \sqrt{|\mathcal{S}_m|}\varepsilon$, $\mathrm{UCB}_m(a) \geq \mu_a - \sqrt{|\mathcal{S}_m|}\varepsilon$. Moreover, as $\mathrm{UCB}_m(a) = \min\{\mathrm{UCB}_{m,1}(a), \mathrm{UCB}_{m,2}(a)\}$, we know that for all $a \in \mathcal{A}_m$

$$\mathrm{UCB}_{m,1}(a) - 2\mathsf{rad}_{m,a}^N - 2\varepsilon\sqrt{|\mathcal{S}_m|} = \widehat{\mu}_{m,1}(a) - \mathsf{rad}_{m,a}^N - 2\varepsilon\sqrt{|\mathcal{S}_m|} \leq \langle a, \theta\rangle,$$

$$\mathrm{UCB}_{m,2}(a) - 2\mathsf{rad}_{m,a}^F - 2\varepsilon\sqrt{|\mathcal{S}_m|} = \widehat{\mu}_{m,2}(a) - \mathsf{rad}_{m,a}^F - 2\varepsilon\sqrt{|\mathcal{S}_m|} \leq \langle a, \theta\rangle,$$

$$\mathrm{LCB}_m(a) + 2\mathsf{rad}_{m,a}^F + 2\varepsilon\sqrt{|\mathcal{S}_m|} = \widehat{\mu}_{m,2}(a) + \mathsf{rad}_{m,a}^F + 2\varepsilon\sqrt{|\mathcal{S}_m|} \geq \langle a, \theta\rangle.$$

If $B \leq \mu^\star$, then $a^\star \in \mathcal{A}_m$ according to [Lemma B.2](#). Moreover, if $a$ is not eliminated in epoch $m$, we have $\mathrm{UCB}_m(a) + 4\sqrt{|\mathcal{S}_m|}\varepsilon \geq \max\{\mathrm{LCB}_m(a^\star), B\}$, meaning that

$$\langle a, \theta\rangle + 2\mathsf{rad}_{m,a}^F + 2\varepsilon\sqrt{|\mathcal{S}_m|}$$
$$\geq \widehat{\mu}_{m,2}(a) + \mathsf{rad}_{m,a}^F$$
$$\geq \mathrm{UCB}_m(a)$$
$$\geq \max\{\mathrm{LCB}_m(a^\star), B\} - 4\sqrt{|\mathcal{S}_m|}\varepsilon$$
$$\geq \mathrm{LCB}_m(a^\star) - 4\sqrt{|\mathcal{S}_m|}\varepsilon$$
$$= \widehat{\mu}_{m,2}(a^\star) - \mathsf{rad}_{m,a^\star}^F - 4\sqrt{|\mathcal{S}_m|}\varepsilon$$

$$\geq \langle a^\star, \theta \rangle - 2\mathrm{rad}_{m,a^\star}^F - 6\sqrt{|\mathcal{S}_m|}\varepsilon.$$

Since $\mathrm{rad}_{m,a}^F = \sum_{i=1}^{|\mathcal{S}_m|} |\lambda_{m,i}^{(a)}| \cdot \mathrm{rad}_{m,a_{m,i}}^F$ with $\|\lambda_m^{(a)}\|_2 \leq 1$, we have that $\|\lambda_m^{(a)}\|_1 \leq \sqrt{|\mathcal{S}_m|}$ and

$$\Delta_a \leq 4\sqrt{|\mathcal{S}_m|} \left( \max_{a \in S_m} \mathrm{rad}_{m,a}^F + 2\varepsilon \right) = 4\sqrt{3n} \max_{a \in S_m} \mathrm{rad}_{m,a}^F + 8\sqrt{3n}\varepsilon \leq \frac{8\sqrt{n}\beta}{\min_{a' \in \mathcal{S}_m} \sqrt{c_m(a')}} + 16\sqrt{n}\varepsilon.$$

If $B \geq \mu^\star$, then we have

$$\mu^\star \leq B \leq \mathrm{UCB}_m(a) + 4\sqrt{|\mathcal{S}_m|}\varepsilon \leq \mu_a + 2\mathrm{rad}_{m,a}^F + 6\sqrt{|\mathcal{S}_m|}\varepsilon,$$

where the second inequality is because $a$ is not eliminated in epoch $m$. Therefore, we always have

$$\Delta_a \leq 2\mathrm{rad}_{m,a}^F + 6\sqrt{|\mathcal{S}_m|}\varepsilon \leq \frac{8\sqrt{n}\beta}{\min_{a' \in \mathcal{S}_m} \sqrt{c_m(a')}} + 12\sqrt{n}\varepsilon.$$

In addition, we know that for all $a \in \mathcal{S}_m$,

$$2^m = |\mathcal{S}_m| \leq c_m(a) + \frac{D}{|\mathcal{S}_m|} + 1 \leq c_m(a) + \frac{2D}{|\mathcal{S}_m|}.$$

Therefore, if $12\sqrt{n}\varepsilon \geq \frac{\Delta_a}{2}$, then we have

$$2^m \Delta_a \leq 2^m \cdot 24\sqrt{n}\varepsilon;$$

otherwise, we have $\Delta_a \leq \frac{8\sqrt{n}\beta}{\min_{a \in \mathcal{S}_m} \sqrt{c_m(a)}} + 12\sqrt{n}\varepsilon \leq \frac{8\sqrt{n}\beta}{\min_{a \in \mathcal{S}_m} \sqrt{c_m(a)}} + \frac{\Delta_a}{2}$ and

$$\Delta_a \leq \frac{16\sqrt{n}\beta}{\min_{a' \in \mathcal{S}_m} \sqrt{c_m(a')}},$$

and we can obtain that

$$\min_{a' \in S_m} c_m(a') \cdot \Delta_a \leq \frac{256 d\beta^2}{\Delta_a}.$$

Combining the above two cases, we know that for all $a \in \mathcal{A}_m$,

$$2^m \cdot \Delta_a \leq 2^m \cdot 24\sqrt{n}\varepsilon + \min_{a' \in \mathcal{S}_m} c_m(a') \cdot \Delta_a + \frac{2D\Delta_a}{|\mathcal{S}_m|} \leq 2^m \cdot 24\sqrt{n}\varepsilon + \frac{256 n\beta^2}{\Delta_a} + \frac{2D\Delta_a}{|\mathcal{S}_m|}.$$

$\square$

The following lemma is a counterpart of Lemma A.5.

**Lemma B.4.** *Algorithm 4 guarantees that under Event 1, if an action $a$ is eliminated at the end of epoch $m$ (meaning that $a \in \mathcal{A}_m$), then*

$$B \leq \langle a, \theta \rangle + \mathrm{rad}_{m,a}^N + \sum_{i=1}^{|\mathcal{S}_m|} |\lambda_{m,i}^{(a)}| \cdot \left( \frac{2d(a_{m,i})}{2^m |\mathcal{S}_m|} + \frac{16 \log T + 2}{2^m} \right) + 8\sqrt{|\mathcal{S}_m|}\varepsilon,$$

*where $d(a) = D\mu_a$.*

*Proof.* Under Event 1, we know that for all $a \in \mathcal{A}_m$,

$$\langle a, \theta \rangle = \sum_{i=1}^{|\mathcal{S}_m|} \lambda_{m,i}^{(a)} \langle a_{m,i}, \theta^\star \rangle$$

$$\begin{aligned}
&= \sum_{i=1}^{|\mathcal{S}_m|} \lambda_{m,i}^{(a)} (\mu_{a_{m,i}} - \varepsilon_{a_{m,i}}) && \text{(since } \mu_a = \langle a, \theta^\star \rangle + \varepsilon_a) \\
&\geq \sum_{i=1}^{|\mathcal{S}_m|} \lambda_{m,i}^{(a)} \cdot \mu_{a_{m,i}} - \sqrt{|\mathcal{S}_m|}\varepsilon && \text{(since } \|\lambda_m^{(a)}\|_1 \leq \sqrt{|\mathcal{S}_m|}) \\
&\geq \sum_{i=1}^{|\mathcal{S}_m|} \lambda_{m,i}^{(a)} \cdot \widehat{\mu}_m(a_{m,i}) - \mathsf{rad}_{m,a}^N - 3\sqrt{|\mathcal{S}_m|}\varepsilon && \text{(since Event 1 holds)} \\
&\geq \sum_{i=1}^{|\mathcal{S}_m|} \lambda_{m,i}^{(a)} \cdot \widehat{\mu}_m^{sgn(\lambda_{m,i}^{(a)})}(a_{m,i}) - \mathsf{rad}_{m,a}^N - \sum_{i=1}^{|\mathcal{S}_m|} |\lambda_{m,i}^{(a)}| \cdot \frac{|E_m(a_{m,i})|}{2^m} - 3\sqrt{|\mathcal{S}_m|}\varepsilon && \text{(using Eq. (35) and Eq. (36))} \\
&= \mathrm{UCB}_{m,1}(a) - \mathsf{rad}_{m,a}^N - \sum_{i=1}^{|\mathcal{S}_m|} |\lambda_{m,i}^{(a)}| \cdot \frac{|E_m(a_{m,i})|}{2^m} - 3\sqrt{|\mathcal{S}_m|}\varepsilon \\
&\geq \mathrm{UCB}_{m,1}(a) - \mathsf{rad}_{m,a}^N - \sum_{i=1}^{|\mathcal{S}_m|} |\lambda_{m,i}^{(a)}| \cdot \left( \frac{2d(a_{m,i})}{2^m |\mathcal{S}_m|} + \frac{16 \log KT + 2}{2^m} \right) - 4\sqrt{|\mathcal{S}_m|}\varepsilon. && \text{(since Event 1 holds)}
\end{aligned}$$

Since $\mathrm{UCB}_{m,1}(a) \geq B - 4\sqrt{|\mathcal{S}_m|}\varepsilon$ (as $a$ is not eliminated at the end of epoch $m$), we have

$$B \leq \langle a, \theta \rangle + \mathsf{rad}_{m,a}^N + \sum_{i=1}^{|\mathcal{S}_m|} |\lambda_{m,i}^{(a)}| \cdot \left( \frac{2d(a_{m,i})}{2^m |\mathcal{S}_m|} + \frac{16 \log T + 2}{2^m} \right) + 8\sqrt{|\mathcal{S}_m|}\varepsilon.$$

$\square$

The following lemma is a counterpart of Lemma A.6.

**Lemma B.5.** *If $B \geq \frac{\mu^\star}{2}$ and Event 1 holds, Algorithm 4 guarantees that if $a$ is not eliminated at the end of epoch $m$, then we also have*

$$2^m \Delta_a \leq \frac{256 n \beta^2}{\Delta_a} + \frac{12 \sum_{i=1}^{|\mathcal{S}_m|} |\lambda_{m,i}^{(a)}| \cdot d(a_{m,i})}{|\mathcal{S}_m|} + (128 \log T + 16)\sqrt{n} + 2^m \cdot 64\sqrt{n}\varepsilon,$$

*where $d(a) = D\mu_a$.*

*Proof.* If $\langle a, \theta \rangle \geq \frac{B}{2}$, we know that $\Delta_a = \langle a^\star - a, \theta \rangle \leq 3 \langle a, \theta \rangle$. Using Lemma B.3, we can obtain that

$$\begin{aligned}
2^m \cdot \Delta_a &\leq 2^m \cdot 24\sqrt{n}\varepsilon + \frac{256 n \beta^2}{\Delta_a} + \frac{2D \langle a, \theta \rangle}{|\mathcal{S}_m|} \\
&\leq 2^m \cdot 24\sqrt{n}\varepsilon + \frac{256 n \beta^2}{\Delta_a} + \frac{2 \sum_{i=1}^{|\mathcal{S}_m|} |\lambda_{m,i}^{(a)}| \cdot d(a_{m,i})}{|\mathcal{S}_m|}.
\end{aligned}$$

If $\langle a, \theta \rangle \leq \frac{B}{2}$, we have $3(B - \langle a, \theta \rangle) \geq \frac{3B}{2} \geq \langle a^\star - a, \theta \rangle$. Using Lemma A.5, we know that

$$\Delta_a \leq \mu_a \leq \underbrace{3 \cdot \mathsf{rad}_{m,a}^N}_{\texttt{Term (1)}} + \underbrace{3 \sum_{i=1}^{|\mathcal{S}_m|} |\lambda_{m,i}^{(a)}| \cdot \left( \frac{2d(a_{m,i})}{2^m |\mathcal{S}_m|} + \frac{16 \log T + 2}{2^m} \right) + 24\sqrt{|\mathcal{S}_m|}\varepsilon}_{\texttt{Term (2)}}.$$

If $\texttt{Term (1)} \geq \texttt{Term (2)}$, we have

$$\Delta_a \leq \mu_a \leq 6\mathsf{rad}_{m,a}^N \varepsilon \leq 6\sqrt{|\mathcal{S}_m|} \max_{a_m \in \mathcal{S}_m} \mathsf{rad}_{m,a_m}^N \leq \frac{12\beta\sqrt{n}}{2^{m/2}},$$

meaning that $2^m \Delta_a \leq \frac{144 n \beta^2}{\Delta_a}$. Otherwise, we have

$$\Delta_a \leq 6 \sum_{i=1}^{|\mathcal{S}_m|} |\lambda_{m,i}^{(a)}| \cdot \left( \frac{2d(a_{m,i})}{2^m |\mathcal{S}_m|} + \frac{16 \log T + 2}{2^m} \right) + 96\sqrt{n}\varepsilon,$$

meaning that

$$2^m \Delta_a \leq \frac{12 \sum_{i=1}^{|\mathcal{S}_m|} |\lambda_{m,i}^{(a)}| \cdot d(a_{m,i})}{|\mathcal{S}_m|} + (96 \log T + 12)\sqrt{n} + 2^m \cdot 96\sqrt{n}\varepsilon.$$

Combining both cases, we know that

$$2^m \Delta_a \leq \frac{256 n \beta^2}{\Delta_a} + \frac{12 \sum_{i=1}^{|\mathcal{S}_m|} |\lambda_{m,i}^{(a)}| \cdot d(a_{m,i})}{|\mathcal{S}_m|} + (96 \log T + 12)\sqrt{n} + 2^m \cdot 96\sqrt{n}\varepsilon.$$

$\square$

Now we are ready to prove our main result Theorem B.1.

*Proof of Theorem B.1.* Combining Lemma B.3 and Lemma B.5 and following the exact same process of obtaining Eq. (37) in Theorem A.1, we can obtain that for a fixed value of $B$, Algorithm 4 guarantees that

$$\text{Reg}_B \leq \mathcal{O}\left( \min\left\{ \frac{n^2 \log(KT) \log(T/n)}{\Delta_{\min}}, n\sqrt{|\mathcal{T}_B| \log(KT)} \right\} + \varepsilon\sqrt{n}|\mathcal{T}_B| \right)$$

$$+ \sum_{m=1}^{\lceil \log_2(|\mathcal{T}_B|/3n) \rceil} \sum_{a \in \mathcal{S}_m} \mathcal{O}\left( \min\left\{ \frac{\sum_{i=1}^{|\mathcal{S}_{m-1}|} |\lambda_{m-1,i}^{(a)}| \cdot d(a_{m-1,i})}{n}, \frac{D\Delta_a}{n} \right\} \right)$$

$$\leq \mathcal{O}\left( \min\left\{ \frac{n^2 \log(KT) \log(T/n)}{\Delta_{\min}}, n\sqrt{|\mathcal{T}_B| \log(KT)} \right\} + \varepsilon\sqrt{n}|\mathcal{T}_B| \right.$$

$$\left. + \min\left\{ \sum_{m=1}^{\lceil \log_2(|\mathcal{T}_B|/3n) \rceil} \sum_{i=1}^{|\mathcal{S}_{m-1}|} d(a_{m-1,i}), D\Delta_{\max} \log(T/n) \right\} \right).$$

According to Lemma B.2, there are at most $\lceil \log_2(1/\mu^\star) \rceil$ different values of $B$. With an abuse of notation, we define $\mathcal{S}_m^{(B)} = \{a_{m,i}^{(B)}\}_{i \in [3n]}$ to be the volumetric spanner at epoch $m$ with the reward guess $B$. Taking summation over all these values, we can obtain that

$$\text{Reg} \leq \mathcal{O}\left( \min\left\{ \frac{n^2 \log(KT) \log(T/n) \log(1/\mu^\star)}{\Delta_{\min}}, n\sqrt{T \log(KT) \log(1/\mu^\star)} \right\} + \varepsilon\sqrt{n}T \right)$$

$$+ \mathcal{O}\left( \min\left\{ \sum_{j=0}^{\lceil \log_2(1/\mu^\star) \rceil} \sum_{i=1}^{3n} d(a_{m-1,i}^{(2^{-j})}), D\Delta_{\max} \log(1/\mu^\star) \log(T/n) \right\} \right),$$

completing the proof. $\square$

While we can further apply a similar analysis to the one in Theorem A.1 to bound the term $\sum_{j=0}^{\lceil \log_2(1/\mu^\star) \rceil} \sum_{i=1}^{3n} d(a_{m-1,i}^{(2^{-j})})$ and obtain a bound with respect to $d^\star$, since $d^\star \geq D\Delta_{\max} + \varepsilon$, this $d^\star$ dependent bound does not provide a significantly better regret guarantee in the worst case. This difference in loss versus reward is also pointed out in (Schlisselberg et al., 2025) in the MAB setting. We keep this term in the upper bound since this quantity can still be potentially smaller than $D\Delta_{\max} \log(1/\mu^\star) \log(T/n)$.

# C. Omitted Details in Section 4

In this section, we provide the omitted details in Section 4. We start with the following lemma that is a standard application of the Azuma-Hoeffding's inequality.

**Lemma C.1** (Proposition 2 in (Hanna et al., 2023))**.** *For each epoch $m$, Algorithm 2 guarantees that with probability at least $1 - \frac{\delta}{T}$, the following holds:*

$$\left| \langle g(\theta), \theta' \rangle - \left\langle g^{(m)}(\theta), \theta' \right\rangle \right| \leq 2\sqrt{\frac{\log(2T|\Theta'|/\delta)}{2^{m-1}}}, \ \ \forall \theta, \theta' \in \Theta'.$$

Next, we provide the proof for Theorem 4.1.

*Proof of Theorem 4.1.* Define $\theta_0 = \operatorname{argmin}_{\theta' \in \Theta'} \|\theta' - \theta\|_2$. Following the analysis of Hanna et al. (2023), we decompose the regret $\operatorname{Reg}_m$ within epoch $m$ as follows:

$$
\begin{aligned}
\operatorname{Reg}_m &= \mathbb{E}\left[ \sum_{\tau=2^{m-1}+1}^{2^m} \left( \left\langle \operatorname*{argmin}_{a \in \mathcal{A}_t} \langle a, \theta_t \rangle, \theta \right\rangle - \min_{a_\tau^\star \in \mathcal{A}_\tau} \langle a_\tau^\star, \theta \rangle \right) \right] \\
&\leq \mathbb{E}\left[ \sum_{\tau=2^{m-1}+1}^{2^m} \left( \left\langle \operatorname*{argmin}_{a \in \mathcal{A}_t} \langle a, \theta_t \rangle, \theta_0 \right\rangle - \min_{a_\tau^\star \in \mathcal{A}_\tau} \langle a_\tau^\star, \theta_0 \rangle \right) \right] + \mathcal{O}\left( \frac{2^{m-1}}{T} \right) \\
&= \mathbb{E}\left[ \sum_{\tau=2^{m-1}+1}^{2^m} \langle g(\theta_t) - g(\theta_0), \theta_0 \rangle \right] + \mathcal{O}\left( \frac{2^{m-1}}{T} \right) \\
&= \underbrace{\mathbb{E}\left[ \sum_{\tau=2^{m-1}+1}^{2^m} \left\langle g(\theta_t) - g^{(m)}(\theta_t), \theta_0 \right\rangle \right]}_{\text{ERR-TERM}(1)} + \underbrace{\mathbb{E}\left[ \sum_{\tau=2^{m-1}+1}^{2^m} \left\langle g^{(m)}(\theta_t) - g^{(m)}(\theta_0), \theta_0 \right\rangle \right]}_{\text{REG-NCTX}} \\
&\quad + \underbrace{\mathbb{E}\left[ \sum_{\tau=2^{m-1}+1}^{2^m} \left\langle g^{(m)}(\theta_0) - g(\theta_0), \theta_0 \right\rangle \right]}_{\text{ERR-TERM}(2)} + \mathcal{O}\left( \frac{2^{m-1}}{T} \right),
\end{aligned}
$$

where the second equality is because $\mathbb{E}\left[\min_{a \in \mathcal{A}_t} \langle a, \theta_0 \rangle \right] = \mathbb{E}\left[ \left\langle \operatorname{argmin}_{a \in \mathcal{A}_t} \langle a, \theta_0 \rangle, \theta_0 \right\rangle \right] = \langle g(\theta_0), \theta_0 \rangle$.

For ERR-TERM(1) and ERR-TERM(2), we apply Lemma C.1 to bound both terms by $\mathcal{O}\left( \sqrt{2^m \log(T|\Theta'|)} \right)$. As for REG-NCTX, this is in fact the regret of misspecified non-contextual linear bandits with action set $\mathcal{X}_m$ and misspecification level $\max_{\theta' \in \Theta'} \left| \langle g^{(m)}(\theta') - g(\theta'), \theta \rangle \right|$, since $\mathbb{E}[u_t] = \langle g(\theta_t), \theta \rangle$ for all $t$. Applying Lemma C.1 again, we know that the misspecification is of order $\varepsilon_m = \mathcal{O}(\sqrt{\log(T|\Theta'|)/2^m})$ with probability at least $1 - \frac{1}{T^2}$. Then, applying the regret guarantee of Algorithm 3 proven in Theorem A.1, we know that

$$\text{REG-NCTX} \leq \mathcal{O}\left( \sqrt{n2^m \log(T|\Theta'|)} \right) + \mathcal{O}\left( \min\{V_{m,1}, V_{m,2}\}, \log(\overline{d}^\star) \min\{W_{m,1}, W_{m,2}\} \right),$$

where $V_{m,1} = \frac{n^2 \log(T|\Theta'|) \log(T/n) \log(\overline{d}^\star)}{\Delta_{\min}^{\text{n-ctx}}}$, $V_{m,2} = n\sqrt{2^m \log(\overline{d}^\star) \log(T|\Theta'|)}$, $W_{m,1} = n\overline{d}^\star \log(T/n) + D\Delta_{\max}^{\text{n-ctx}}$, and $W_{m,2} = D\Delta_{\max}^{\text{n-ctx}} \log(T/n)$. Taking a summation over all $m \in [\lceil \log_2(T) \rceil]$ epochs and using the fact that $|\Theta'| \leq \mathcal{O}(T^n)$ finishes the proof. $\square$

## D. Omitted Details in Section 5

For completeness, we include the pseudo code for the benchmark used in our experiment, that is, `LinUCB` using only the observed feedback; see Algorithm 5.

---

**Algorithm 5:** LinUCB with Delayed Feedback

---

Input: action set $\mathcal{A}$, a parameter $\lambda > 0$.

Initialize: $\widehat{\theta}_1$ arbitrarily, $\beta_t = \sqrt{\lambda} + \sqrt{2 \log T + n \log(1 + \frac{t}{n\lambda})}$ for all $t \in [T]$, $H_1 = \lambda I$.

**for** $t = 1, 2, \ldots, T$ **do**

  Pick

$$a_t = \begin{cases} \operatorname{argmin}_{a \in \mathcal{A}} \left\langle a, \widehat{\theta}_t \right\rangle - \beta \|a\|_{H_t}^{-1}, & \text{in the loss case,} \\ \operatorname{argmax}_{a \in \mathcal{A}} \left\langle a, \widehat{\theta}_t \right\rangle + \beta \|a\|_{H_t}^{-1}, & \text{in the reward case.} \end{cases}$$

  Observe the payoff $u_\tau$ for all $\tau$ such that $\tau + d_\tau \in (t-1, t]$.

  Update $H_{t+1} = H_t + \sum_{\tau : \tau + d_\tau \in (t-1, t]} a_\tau a_\tau^\top$ and $\widehat{\theta}_{t+1} = H_{t+1}^{-1} \sum_{\tau : \tau + d_\tau \leq t} a_\tau u_\tau$.

---

