# OpenReview forum: "Contextual Linear Bandits with Delay as Payoff"
_ICML.cc/2025/Conference — ICML 2025 poster_

### Official Review · Reviewer_AjsP · 2025-03-04

**Overall Recommendation:** 2

**Summary:**

This paper investigates contextual linear bandits in which the payoff (loss or reward) is observed after a delay proportional to the payoff itself. This extends prior research on multi-armed bandits (MAB) with payoff-dependent delays. The authors propose a phased arm elimination algorithm for the non-contextual setting, which selects actions from a *volumetric spanner* of the action set.


## update after rebuttal
After reviewing the other reviewers' comments and the authors' responses, some of my initial concerns have been adequately addressed. I also note that prior work, such as "Multi-player Multi-armed Bandits with Delayed Feedback," considers sub-Gaussian delays. Given this, if extending the noise model to sub-Gaussian distributions is straightforward, I recommend incorporating this improvement in a subsequent version of the paper.

**Claims And Evidence:**

Yes

**Essential References Not Discussed:**

No

**Experimental Designs Or Analyses:**

No

**Methods And Evaluation Criteria:**

Yes

**Other Comments Or Suggestions:**

No.

**Other Strengths And Weaknesses:**

**Strengths**:
This paper extends the understudied delay-as-payoff model from MAB to linear bandits. The work is well-structured, featuring a clear problem formulation, detailed algorithm pseudocode, and concise proof sketches.

**Weaknesses**:
The reliance on mature techniques, such as the spanner method and phased arm elimination, limits the novelty of the proposed approach. While the application of these techniques to the delay-as-payoff model is commendable, the paper does not introduce fundamentally new methodologies.

**Questions For Authors:**

1. The algorithm requires actions and parameters to lie in $\mathbb{R}^n_+$. How critical is this restriction?

2. About the mentioned challenges: The linear form of Eq. (2) remains an efficient LCB for linear bandits. Specifically, at round $t$, let $\hat{\theta}_t$ denote the estimator and $V_t$ represent the covariance matrix of historically selected arms. For any arm $a\in\mathcal{A}$, its estimated reward is given by:

   $$
   \langle a, \hat\{\theta\}\_t \rangle = a \cdot V_t^{-1} \cdot \left( \sum_{\tau=1}^{t-1} u_\tau \cdot a_\tau \right) = a \cdot V_t^{-1} \cdot \left( w_t(\mathcal{A} \setminus \{a\}) + a \cdot R_t(a) \right),
   $$
   where $w_t(\mathcal{A} \setminus \{a\})$ is the weighted sum of historically selected arms excluding $a$, and $R_t(a)$ is the cumulative reward of arm $a$. Without loss of generality, we assume the rewards in $w_t(\mathcal{A} \setminus \{a\})$ are observed.

   Since $V_t$ is positive definite, $a \cdot V_t^{-1} a \geq 0$. Thus, for arm $a$, setting its unobserved rewards to zero yields a lower bound compared to the observed case.

   For another arm $a'\neq a$, the efficiency of the LCB requires $a' \cdot V_t^{-1} a \geq 0$. Given that $\mathcal{A} \subseteq \mathbb{R}^n_+$, the inner product of any two arms is non-negative, ensuring $a' \cdot V_t^{-1} a \geq 0$ holds.

   This raises the question: Is the spanner technique truly necessary?

**Relation To Broader Scientific Literature:**

This paper extends the understudied delay-as-payoff model from MAB to linear bandits.

**Theoretical Claims:**

No

---

> ### Author Rebuttal · Authors · 2025-03-31
>
> Thanks for your valuable comments. We address the issues mentioned in your review below.
>
> - **Q1: The reliance on mature techniques, such as the spanner method and phased arm elimination, limits the novelty of the proposed approach.**
>
> While we agree that neither volumetric spanner or phased arm elimination is new, combining them to solve the issues that other standard ideas for linear bandits do not seem to be able to solve in our model is a significant contribution in our opinion. In fact, we are not aware of algorithms using similar ideas even for standard linear bandit models.
>
>
> - **Q2: The algorithm requires actions and parameters to lie in $\mathbb{R}^n_+$.**
>
> As mentioned in Footnote 1, we enforce both the action $a$ and the underlying parameter $\theta$ to lie in $\mathbb{R}^n_+$ in order to make sure that the payoff $\langle a,\theta\rangle$, and hence the delay, is non-negative. It is just an important restriction to properly define our model, instead of an assumption to make the analysis work.
>
> - **Q3: An alternative way of estimating LCB of action $a$ using $V_t^{-1} = (\sum_{\tau=1}^ta_{\tau}a_{\tau}^\top)^{-1}$.**
>
> There is a critical issue in your proposal and reasoning. Specifically, you argued that for $a'\neq a$, we have $a^\top V_t^{-1} a' \geq 0$ since all actions are in $\mathbb{R}^n_+$. This is just simply wrong since $V_t^{-1}$ can have negative entries even though $V_t$ does not (for a simple example, consider $V_t=a_1a_1^\top+a_2a_2^\top$ where $a_1=[1;1]$ and $a_2=[1;0]$; then direct calculation shows that $V_t^{-1}=[1,-1;-1,2]$ and $a_2^\top V_t^{-1}a_3= -1<0$ for $a_3=[0;1]$). Therefore, ignoring these actions as you proposed does not lead to an LCB. We hope that this convinces the reviewer that constructing LCB in our problem is nontrivial and that there is clear novelty in our spanner-based approach.

---

> > ### Comment · Reviewer_AjsP · 2025-04-04
> >
> > Thank you for your response. The assumption that the inherent parameters are strictly positive is a strict condition that may not hold in practice, as these parameters are often unobservable for a given problem.

---

> > > ### Author Response · Authors · 2025-04-04
> > >
> > > We thank the reviewer for the further response. We clarify that we in fact only require the realized payoff to be non-negative so that the delay is well-defined. For simplicity, we do so by enforcing parameters $\theta$ and actions $a$ to lie in $\mathbb{R}_+^n$, but again, all our results and analyses naturally follow as long as the realized payoff is non-negative. We believe this is both natural and practical.
> > >
> > > Also, if our response to Q3 addresses your concern about the necessity of the spanner technique, please kindly acknowledge that (so other reviewers know there is no trivial solutions to our problem) and consider re-evaluating the technical strength of our paper. We thank the reviewer again for your time and effort.

---

### Official Review · Reviewer_ddMd · 2025-03-06

**Overall Recommendation:** 2

**Summary:**

This paper studies a contextual linear bandit setting where the reward/loss is delayed by a length of time proportional to the realised reward/loss. For this problem, the authors propose an arm elimination strategy and analyse the regret (including delay penalty) of the proposed algorithm. Experiments in a simulated environment are also provided to complement the theoretical results.

**Claims And Evidence:**

In line 156 (2nd column) it is claimed that it is hard to construct a lower bound equivalent to eq2. It is not clear why this is, and the evidence is a bit vague and not convincing. In particular, it is not clear why we cannot minimize over an appropriately defined confidence set.

**Essential References Not Discussed:**

See above

**Experimental Designs Or Analyses:**

See above (methods section). The experiments also consider a fairly small number of arms and only one sort of reward distribution.

**Methods And Evaluation Criteria:**

Would potentially be good to see experimental comparison to other work on delayed feedback in linear bandits, even if that work does not consider the delay as payoff setting.

**Other Comments Or Suggestions:**

na

**Other Strengths And Weaknesses:**

I question the optimality of the D\Delta_max terms in the regret penalty. In the case where the delays are independent of the reward, the delay penalty can be reduced to the expected delay, why can we not get some sort of average appearing here? In particular, I imagine this average delay will relate to the average reward of the algorithm which in turn could be related to the regret plus average delay of the optimal algorithm. I would find these sorts of results much more insightful. If this cannot be done, it would be interesting to see a lower bound to show that this maximal delay penalty is unavoidable.

As it is, I don't really find the results in this paper surprising given prior results in the MAB setting. The extension from linear bandits to contextual linear bandits seems to follow from a reduction in prior work. I therefore wonder whether there is enough novelty.

On the positive side, I think the algorithm design is interesting, and it is pleasing to see that it works empirically.

**Questions For Authors:**

1)	Can the results be extended to capture more realistic noise models?
2)	Can the dependence on the maximum delay be reduced to something more reasonable and more interpretable?

**Relation To Broader Scientific Literature:**

There is some missing work on delayed feedback in (generalized) linear bandits missing, e.g. the below.

Yang, Y., Zhong, H., Wu, T., Liu, B., Wang, L., & Du, S. S. (2023). A reduction-based framework for sequential decision making with delayed feedback. Advances in Neural Information Processing Systems.

Howson, B., Pike-Burke, C., & Filippi, S. (2023, April). Delayed feedback in generalised linear bandits revisited. In International Conference on Artificial Intelligence and Statistics.

**Theoretical Claims:**

The assumption of bounded noise seems very strong. In particular, it excludes the common linear-Gaussian regression setting. Can the results be relaxed to hold for sub-Gaussian rewards (this may require replacing the maximal delay scaling with some sort of average which I also think would help strength and interpretability of the results, see below).

The proof sketch in section 3.1 seems reasonable to me. The only question I have about the proof is how lamba_{mi} being random effects the analysis?

---

> ### Author Rebuttal · Authors · 2025-03-31
>
> Thanks for your valuable comments. We address the issues mentioned in your review.
>
> - **Q1: It is not clear why it is hard to construct a lower bound equivalent to Eq.(2). In particular, why we cannot minimize over an appropriately defined confidence set.**
>
> We emphasize again the difficulty of obtaining a lower bound similar to Eq.(2) using classic LinUCB estimator ~$\hat{\theta}_{t}$~. When the delay is payoff-independent, Theorem 1 of [Vernade et al., 2020] indeed shows that certain construction of ~$\hat{\theta}_{t}$~ ensures that the norm of ~$\hat{\theta}_{t} - \theta$~ under $V_t^{-1}$ is controlled (where $V_t\approx\sum_{\tau=1}^ta_ta_t^\top$). However, their proof heavily relies on the independence between the delay and payoff. In our model, the payoff and the delay are dependent, and we do not see a way to construct or even approximate a similar confidence set, so we resolve this issue by proposing a novel arm-elimination algorithm that constructs UCB/LCB based on volumetric-spanners.
>
> - **Q2: Can the results be relaxed to hold for sub-Gaussian rewards? Can the results be extended to capture more realistic noise models?**
>
> Since delay is essentially the same as payoff in our model, it does not make sense to have negative payoff, which is why we only consider bounded noise. In other words, allowing sub-Gaussian rewards would require a different delay model, which might be an interesting future direction.
>
> - **Q3: How $\lambda_{m,i}^{(a)}$ being random affects the analysis?**
>
> We are not sure we fully understand this question. Yes, the coefficients $\lambda_{m,i}^{(a)}$ are random, but this does not really introduce any complication to the analysis, and we only use the property $||\lambda_{m}^{(a)}||_2\leq 1$ to control the scale of the confidence range.
>
> - **Q4: More experiment settings.**
>
> Following your (and Reviewer B3d9's) suggestion, we further tested our algorithm in a setting with a larger $K=70$ and $u_t$ drawn from the beta distribution with $\alpha=\mu_{a_t}$ and $\beta=1-\mu_{a_t}$. Moreover, we also added two baselines: OTFLinUCB and OTFLinTS in [Vernade et al., 2019], which are linear bandit algorithms with payoff-independent delay. The results (presented in https://anonymous.4open.science/r/PaperID-8852) show that our algorithm consistently outperforms all baselines in both the payoff-as-reward and payoff-as-loss settings.
>
> - **Q5: The optimality of the $D\Delta_{\max}$ terms in the regret penalty. In the case where the delays are independent of the reward, the delay penalty can be reduced to the expected delay, why can we not get some sort of average appearing here? Can the dependence on the maximum delay be reduced to something more reasonable and more interpretable?**
>
> Note that while $D$ represents the maximum per-round delay, the term $D\Delta_{\max} = \max_{a}D\mu_a - \min_aD\mu_a$ by definition is the gap of the **expected delay** between the best and the worst arms. Therefore, $D\Delta_{\max}$ already represents some kind of ``expected delay'' mentioned by the reviewer. Whether the exact term $D\Delta_{\max}$ is necessary in the regret bound is indeed unclear though.
>
> Thanks for the additional references you suggest and we will incorporate these in our next revision.

---

> > ### Comment · Reviewer_ddMd · 2025-04-02
> >
> > Thanks to the authors for addressing some of my question - in particular the interpretation of my question about lambda was correct. However, I am still concerned about the noise assumptions. I agree with the authors that we do not want negative delays. However, the fact that their model means that they necessarily cannot consider Gaussian noise (which is the most common example of noise in linear bandits) is concerning. This therefore suggests the model is perhaps not realistic enough/not capturing the problem completely. Therefore, I will keep my rating the same.

---

> > > ### Author Response · Authors · 2025-04-02
> > >
> > > We thank the reviewer for elaborating on this concern, but we have to respectfully disagree with the statement that our model is not realistic enough. The fact that sub-Gaussian noise is common in the linear bandit literature does not mean that it is always the most realistic assumption. Indeed, in the applications that we care about, such as those in clinical studies and online advertising (see 2nd paragraph of our introduction or our response to Reviewers PfQm and B3d9), delay and payoff are basically the same thing, so given that the reviewer agrees that negative delay does not make sense, it is clear that sub-Gaussian noise also does not make sense here.
> > >
> > > Moreover, several prior works also consider bounded payoff models in the delayed feedback setting, both in stochastic and adversarial environments (e.g., Vernade et al., 2020; Ito et al., 2020; Van Der Hoeven et al., 2023). Thus, we believe that our modeling choice is not only realistic for many important applications but also aligned with the existing literature on bandits with delay feedback.
> > >
> > > [Ito et al., 2020]: Delay and Cooperation in Nonstochastic Linear Bandits, NeurIPS 2020
> > >
> > > [Vernade et al., 2020]: Linear Bandits with Stochastic Delayed Feedback, ICML 2020
> > >
> > > [Van Der Hoeven et al., 2023]: A Unified Analysis of Nonstochastic Delayed Feedback for Combinatorial Semi-Bandits, Linear Bandits, and MDPs, COLT 2023

---

### Official Review · Reviewer_B3d9 · 2025-03-13

**Overall Recommendation:** 4

**Summary:**

The paper extends the delay-as-payoff model (Schlisselberg et al., 2024) from standard multi-armed bandits (MABs) to contextual linear bandits. This setup arises in practical situations such as clinical trials and modeling time-to-event data for other medical procedures, advertising, wherein the delay in observing a reward (or loss) actually depends on the reward (or loss) itself. More, specifically, the authors assume that the delay at time $t$ is $d_t = D \cdot u_t$, where $u_t$ is the loss at time $t$ and $D>0$ is the maximum possible delay.  The paper considers two set ups: 1) non-contextual linear bandits where the decision-maker selects actions from a fixed action set, 2) contextual linear bandits, where the actions are not fixed but drawn from an unknown distribution $\mathcal{P}$. The objective is to minimize expected pseudo regret. In order to achieve this objective, the authors propose a volumetric spanner based phased elimination method instead of using standard confidence ellipsoid based methods such as LinUCB. They provide an instance-dependent regret bound for non-contextual bandits which matches the standard linUCB regret bound in the no-delay case, and extend the analysis to the contextual linear bandits by adopting the results from Hanna et al., 2023. Their experimental results show that their approach outperforms LinUCB in both delay-as-loss and delay-as-reward settings.

**Claims And Evidence:**

The paper provides the first regret guarantee for contextual linear bandits with payoff-dependent delays. The regret bound for the phased elimination algorithm in the non-contextual linear bandit setting is similar to the no-delay cases and aligns with previous lower bounds in delayed bandit settings. This bound ensures that delay does not significantly increase regret when the optimal action has a small loss. The authors extend their approach to contextual bandits with time-varying action sets. The authors adapt the Hanna et al. (2023) reduction technique to handle varying action sets in contextual bandits. This allows them to use their non-contextual algorithm as a subroutine in the contextual case. This extension provides the first regret guarantee for contextual linear bandits with payoff-dependent delays.  The authors test their algorithm on synthetic linear bandit instances. The results show that their approach outperforms LinUCB in both delay-as-loss and delay-as-reward settings. In Proposition 3.2, the authors state the theoretical complexity of volumetric spanner $\mathcal{S}$ of $\mathcal{A}$ with $|\mathcal{S}| = 3n$ within $O(K n^3 \log{n})$, which is high for large action sets (K Is large) and since $\mathcal{S} \subset \mathcal{A}$, for large $K$, $n$ would potentially also be large, thus making it computationally demanding. There is no discussion of the scalability of this approach for when $n$ (and $K$) is large. The terms $W_1$ and $W_2$ in the regret bound in Theorem 3.3 depend on the expected delay of the optimal action, $d^*$, and  the maximum possible delay, $D$ but scaled by $\Delta_\text{max}$. Thus if all actions have different losses and large losses, the performance could degrade. Although the paper varies the delay structure to check how the algorithm handles different delay settings, it would be nice to see how the algorithm performs under extreme delay settings, which are common in applications such as clinical trials.

**Essential References Not Discussed:**

Phased elimination is used a lot in the best arm identification literature and there is also a lot of work on delayed rewards in that realm that might be cited, such as Grover et al 2018, and there is plenty of other literature but I am not aware of one that studies reward dependent delays in contextual bandits setup.




*Grover, Aditya, et al. "Best arm identification in multi-armed bandits with delayed feedback." International conference on artificial intelligence and statistics. PMLR, 2018.*

**Experimental Designs Or Analyses:**

The paper compares the proposed phased elimination algorithm against LinUCB, a standard method for contextual bandits. The delay structure is systematically varied, allowing for a controlled evaluation of how delay impacts regret. The experiments test both delay-as-loss and delay-as-reward settings, covering different practical scenarios.

**Methods And Evaluation Criteria:**

The use of volumetric spanners is a new exploration technique that reduces the need for explicit parameter estimation. The phased elimination strategy is well-motivated and theoretically justified, and is commonly used in the linear bandits literature. The primary evaluation metric is cumulative regret, which is standard in bandit literature.
It is not clear as to how well the proposed methodology would generalize to cases where one might have stochastic delays or more complex delay scenarios, such as one would expect in real life. Also, the method only compares against LinUCB but there is a vast literature on delayed bandits (reward independent ones) and also modifications to the UCB type algorithms to cater to delayed feedback in contextual bandits framework, such as Zhou et al (2019),  Vernade et al (2020), Lancewicki et al (2021), Vakili et al (2023).


*Zhou, Zhengyuan, Renyuan Xu, and Jose Blanchet. "Learning in generalized linear contextual bandits with stochastic delays." Advances in Neural Information Processing Systems 32 (2019).*

*Vernade, Claire, et al. "Linear bandits with stochastic delayed feedback." International Conference on Machine Learning. PMLR, 2020.*

*Lancewicki, Tal, et al. "Stochastic multi-armed bandits with unrestricted delay distributions." International Conference on Machine Learning. PMLR, 2021.*


*Vakili, Sattar, et al. "Delayed feedback in kernel bandits." International Conference on Machine Learning. PMLR, 2023.*

**Other Comments Or Suggestions:**

It would be nice to see how robust your proposed algorithm is to delays that may also depend on other factors (like contexts) along with payoffs.

**Other Strengths And Weaknesses:**

Strengths:

- While delayed feedback in bandits has been studied (e.g., Joulani et al., 2013), this paper is one of the first to formalize and analyze payoff-dependent delays in contextual linear bandits.
- Standard delayed bandit methods (e.g., LinUCB) use confidence ellipsoid-based exploration. The paper instead leverages volumetric spanners to reduce exploration complexity and circumvents the challenges in using confidence ellipsoid based approach in the presence of delayed feedback. This work generalizes prior results to  dynamic action sets, which makes it a meaningful extension of bandit theory.
- The delay-as-payoff model is clearly defined, with a rigorous problem setup. The regret analysis follows standard proof techniques in bandit literature, making it easy to follow.

Weaknesses:
- There is no comparison with other delayed bandit methods, even those assuming delays are independent of payoffs. It would be valuable to see how they perform relative to the proposed algorithm. Additionally, a computational complexity comparison in terms of runtime would be beneficial.
- The methods seem to have high computational burden when large action sets are considered, especially in terms of scalability of the volumetric spanner.
- The reduction from contextual linear bandits to non-contextual linear bandits using Hanna et al. (2023) is not very clear and the relationship with an $\epsilon$- misspecified model in not apparent to a reader not familiar with the paper.

**Questions For Authors:**

1) Why consider the expected pseudo regret and not the expected regret itself?
2) In the delay-as-reward setting, it is assumed that the noise $\epsilon_a \leq \epsilon$, which seems to be a strong assumption. Can this be relaxed?
3) How does one check if the delays actually scale with payoffs in real life?
4) Can you take into account contextual information in your framework?

**Relation To Broader Scientific Literature:**

There is a vast literature on bandits with delayed rewards. While a lot of the literature assumes delays as being independent of rewards, arms and contexts, there is a growing interest in studying the more practical setting where delayed feedback could be more complicated, such a arm-dependent delays (Gael et al 2020), heavy tailed delays (Blanchet et al, 2020)  depend on a variety of factors such as payoffs (losses or rewards)   as in this paper and others like Tang et al, 2024 as cited in this paper. In that light, to my knowledge this is the first paper studying delayed rewards dependent on payoffs in a linear contextual bandits setting.





*Gael, Manegueu Anne, et al. "Stochastic bandits with arm-dependent delays." International Conference on Machine Learning. PMLR, 2020.*

*Blanchet, Jose, Renyuan Xu, and Zhengyuan Zhou. "Delay-adaptive learning in generalized linear contextual bandits." Mathematics of Operations Research 49.1 (2024): 326-345.*

*Tang, Yifu, Yingfei Wang, and Zeyu Zheng. "Stochastic multi-armed bandits with strongly reward-dependent delays." International Conference on Artificial Intelligence and Statistics. PMLR, 2024.*

**Theoretical Claims:**

While I haven’t verified every step of the proofs, the proof sketch appears correct and aligns with standard linear bandit theory. The authors do not claim minimax optimality for their results, which presents an interesting open question deserving further analysis.

---

> ### Author Rebuttal · Authors · 2025-03-31
>
> Thanks for your valuable and positive comments and your acknowledgment on our initiation to study linear contextual bandits with payoff-dependent delay. We address the issues mentioned in your review below.
>
> - **Q1: There is no comparison with other delayed linear bandit methods, even those assuming delays are independent of payoffs.**
>
> Following your (and Reviewer ddMd's) suggestion, we conducted extra experiments where we added two other baselines: OTFLinUCB and OTFLinTS in [Vernade et al., 2019], which are linear bandit algorithms with payoff-independent delay. The results (presented in https://anonymous.4open.science/r/PaperID-8852) show that our algorithm consistently outperforms all baselines in both the payoff-as-reward and payoff-as-loss settings.
>
> - **Q2: No discussion of the scalability/computational cost of this approach for when $n$ (and $K$) is large.**
>
> We apologize for not discussing time complexity in the paper. However, we emphasize that our algorithm is in fact **even more computationally efficient** than the classic LinUCB algorithm. More specifically, the time complexity of our algorithm over $T$ rounds is $O(nT+Kn^3\log n\log(T/n))$ in total since we only compute the volumetric spanner at the beginning of each epoch, and the total number of epoch is $\log(T/n)$. On the other hand, LinUCB's time complexity is $O(Kn^2T)$ since computing the UCB of each action requires $O(n^2)$ time. Therefore, our algorithm is in fact more efficient. We will add this discussion in the next revision. Thanks for pointing this out.
>
> - **Q3: Why expected pseudo regret?**
>
> This is standard in the stochastic bandit literature (such as the UCB paper),  especially when the goal is to derive logarithmic regret. This is because if one were to consider expected regret, then even if the algorithm always picks the optimal arm, the derivation in the stochastic losses still contributes to $\sqrt{T}$ regret.
>
> - **Q4: The reduction from contextual linear bandits to non-contextual linear bandits using Hanna et al. (2023) is unclear; assumption on the misspecification level $\epsilon_a\leq \epsilon$.**
>
> Note that the reason we consider the misspecified setting is mostly to enable the use of the contextual-to-non-contextual reduction proposed by [Hanna et al., 2023], and in that reduction, the misspecification level of each arm is indeed **uniformly bounded by a known value**. To be clear, we explain the high-level idea of the reduction and why it requires a misspecifed model below, and we will add more discussion to the paper: at a high level, their reduction treats each model parameter $\theta$ as an action and goes in epochs. Since the distribution of the action set ~$\mathcal{A}_t$~ at each round $t$ is unknown, at epoch $m$, they can only estimate each action $g(\theta) = E_{\mathcal{A} \sim \mathcal{P}} \left[ \arg\min_{a \in \mathcal{A}} \langle a, \theta \rangle \right]$ using historical data (denoted as $g^{(m)}(\theta)$ in Line 1 of Algorithm 2). This leads to a misspecified model since the true expected loss of picking $a_t$ is $\langle g(\theta_t), \theta\rangle$ while the algorithm considers the loss model of $\langle g^{(m)}(\theta_t), \theta\rangle$. According to Lemma C.1, the gap between $\langle g(\theta_t), \theta\rangle$ and $\langle g^{(m)}(\theta_t), \theta\rangle$ is indeed bounded by a known value $O(\sqrt{1/2^m})$, which is the reason why we only require an algorithm with a uniform and known misspecification level.
>
> - **Q5: How does one check if the delays actually scale with payoffs in real life?**
>
> As mentioned in our introduction, for some applications, this is true by definition: in medical domains, the delay in observing progression-free survival (reward) or postoperative length of stay (loss) is exactly the metric itself; in advertising, the delay in observing average time on page (reward) or the time to re-engagement (loss) is also the metric itself. In other applications where this is less obvious, one possibility is to collect historical data and apply a linear regression on delays versus payoff to see if this is a good model.
>
> - **Q6: Can you take into account contextual information in your framework?**
>
> We assume that reviewer is asking whether we can also allow context-dependent delay. However, note that our payoffs depend on the context, so the delay, essentially being the same as the payoff in our model, already depends on the context.
>
> Thanks for pointing out other references. We will incorporate these into our next revision.

---

> > ### Comment · Reviewer_B3d9 · 2025-04-01
> >
> > Thank you for the detailed response. I would like to maintain my score.

---

### Official Review · Reviewer_PfQm · 2025-03-14

**Overall Recommendation:** 3

**Summary:**

The authors try to extend the delay-as-payoff model to contextual linear bandits. The main novelty here is to apply a phased arm elimination procedure by only picking the **volumetric spanners** of the action set in order to handle both payoff-dependent delays and large action sets. Further extension is discussed in the case with varying action sets.

**Claims And Evidence:**

Yes

**Essential References Not Discussed:**

Not aware of any.

**Experimental Designs Or Analyses:**

Yes. I checked the experimental setting, which looks good to me.

**Methods And Evaluation Criteria:**

Yes

**Other Comments Or Suggestions:**

N/A

**Other Strengths And Weaknesses:**

To Note: I am not an expert in the algorithmic theory of bandit analysis. Hence, I will provide more general questions regarding the work and less on the technical side.

Strength: The authors did interesting extensions to the MAB setting for delay as payoff. The phase-elimination trick avoids the difficulty of estimating the LCB in the delayed case and leads to meaningful regret bounds. The technical discussion is quite deep.

Weakness and general questions:
1. I noticed that the authors (as well as earlier literature) used a particular form for modeling the delay as a linear function of the loss. Can this be more general? Say only assume delay is within some function class defined on the loss. It's hard to imagine in practice the delays are always linear dependence. How much does the theory rely on such an assumption?
2. The authors did a great explanation on the derived regret bounds. Is there an optimality result (lower bound) as well?
3. The derived regret bound imposes bounded delay assumptions. In practice, there could be infinite scale of delay, which is actually discussed in paper such as Gael et al (2020). Is there a way to treat for example long delay vs short delay to generalize the analysis?
4. From the simulation results and the discussion around, LinUCB also performs fair as an ad-hoc strategy, especially when $n$ is small. Is there any quantification for the performance of LinUCB to show it is theoretically worse or completely fail in certain delay-as-pay-off setting?

**Questions For Authors:**

N/A

**Relation To Broader Scientific Literature:**

This work broadens the literature of delayed bandits.

**Theoretical Claims:**

I have checked the proof of the main theorem (theorem 3.3), which reads correct to me.

---

> ### Author Rebuttal · Authors · 2025-03-31
>
> Thanks for your valuable comments. We address the issues mentioned in your review below.
>
> - **Q1: Why assume the delay as a linear function of payoff; other general models?**
>
> Our goal is to extend the same delay-as-payoff model of Schlisselberg et al. (2024) from MAB to contextual linear bandits, and as mentioned in the 2nd paragraph of our introduction (as well as in Schlisselberg et al.), there are indeed many applications where this model is valid. For example, in medical domains, the delay in observing progression-free survival (reward) or postoperative length of stay (loss) is exactly the metric itself; in advertising, the delay in observing average time on page (reward) or the time to re-engagement (loss) is also the metric itself. The analysis of both our work and Schlisselberg et al. (2024) does heavily rely on this model. That said, we agree that extending the results to more general models is definitely an interesting direction (as also pointed out in the conclusion of our paper).
> - **Q2: Is there an optimality result (lower bound) as well?**
>
> As also mentioned in the conclusion section, we do not know whether our bounds are optimal and leave this as future directions. In fact, the optimal bounds in the simpler MAB case is also unknown.
> - **Q3: In practice, there could be infinite scale of delay, which is actually discussed in paper such as Gael et al (2020). Is there a way to treat for example long delay vs short delay to generalize the analysis?**
>
> While this is indeed an interesting case, it unfortunately does not fit into the delay-as-payoff model since infinite scale of delay also implies infinite scale of payoff, in which case no sublinear regret should be possible.
>
> - **Q4: Is there any quantification for the performance of LinUCB to show it is theoretically worse or completely fail in certain delay-as-payoff setting?**
>
> To the best of our knowledge, this question is indeed still open even for the simpler MAB setting. Our conjecture is that since these algorithms do not take into account the delay-as-payoff structure, they indeed could completely fail in some worst-case environments.

---

> > ### Comment · Reviewer_PfQm · 2025-04-03
> >
> > Thanks for your responses.
> >
> > Q1 & Q2: I am happy to see the future extension of the work that addresses these questions.
> >
> > Q3: for infinite-scale delays, that is related to my question Q1 regarding the model of the delay-payoff. Also would like to further see the extension here.
> >
> > Q4: Thanks for clarifying this point.
> >
> > In general I can tell that this is a hard yet valuable setup with many open questions. Although I am not an expert in bandit algorithm theory, I do respect the authors' efforts in provide theoretical insights in modeling/analyzing such a setting and will keep my rating as 3.

---

### Decision · Program_Chairs · 2025-05-01

**Decision:**

Accept (poster)

**Comment:**

The paper extends the delay-as-payoff model, from a standard bandit setting to contextual linear bandits. In this setting, the delay of the feedback is linearly dependent on the reward, with applications in medical trials and advertising/recommendation. A novel algorithm is proposed, using a phase-elimination trick and leverages volumetric spanners to avoid the challenges of using confidence ellipsoids in the presence of delayed feedback. While the reviewers agree the setting is technically challenging and novel, there are some concerns regarding some of the assumptions required to obtain the theoretical results: bounded noise, reward scaling linearly as delay etc. and novelty of the approach (phase elimination is prevalent in best-arm identification and reduction of the contextual setting is taken from Hanna et al. 2023).

Nonetheless, this submission presents interesting technical results and represents a promising step towards settings relying on weaker assumptions and therefore a solid contribution to the conference program.

As a final note, perhaps the authors would like to know that delayed bandits with intermediate observations (mentioned in Conclusions) were studied prior to Esposito et al. (https://proceedings.mlr.press/v119/vernade20b.html).